

# Anti-*Acanthamoeba* activity of a semi-synthetic mangostin derivative and its ability in removal of *Acanthamoeba triangularis* WU19001 on contact lens

Julalak Chuprom[1,2], Suthinee Sangkanu[1], Watcharapong Mitsuwan[3], Rachasak Boonhok[4], Wilawan Mahabusarakam[5], L. Ravithej Singh[6,7], Ekachai Dumkliang[8], Kritamorn Jitrangsri[9], Alok K. Paul[10], Sirirat Surinkaew[11], Polrat Wilairatana[12], Maria de Lourdes Pereira[13,14], Mohammed Rahmatullah[15], Christophe Wiart[16], Sonia Marlene Rodrigues Oliveira[13,17] and Veeranoot Nissapatorn[1]

[1] School of Allied Health Sciences, Southeast Asia Water Team (SEA Water Team) and World Union for Herbal Drug Discovery (WUHeDD), Walailak University, Nakhon Si Thammarat, Thailand
[2] School of Languages and General Education (SOLGEN), Walailak University, Nakhon Si Thammarat, Thailand
[3] Akkhraratchakumari Veterinary College, Walailak University, Nakhon Si Thammarat, Thailand
[4] Department of Medical Technology, School of Allied Health Sciences, Research Excellence Center for Innovation and Health Products (RECIHP), Walailak University, Nakhon Si Thammarat, Thailand
[5] Division of Physical Science, Faculty of Science, Prince of Songkla University, Songkhla, Thailand
[6] Fluoro-Agrochemicals Division, CSIR–Indian Institute of Chemical Technology, Tarnaka, Hyderabad, India
[7] Chemical Sciences Division, Academy of Scientific and Innovative Research, Ghaziabad, India
[8] Faculty of Pharmacy, Silpakorn University, Pharmaceutical Development of Green Innovations Group (PDGIG), Nakhon Pathom, Thailand
[9] Department of Pharmaceutical Chemistry, Faculty of Pharmacy, Silpakorn University, Nakhon Pathom, Thailand
[10] School of Pharmacy and Pharmacology, University of Tasmania, Hobart, Australia
[11] Department of Medical Technology, School of Allied Health Sciences, Walailak University, Nakhon Si Thammarat, Thailand
[12] Department of Clinical Tropical Medicine, Faculty of Tropical Medicine, Mahidol University, Bangkok, Thailand
[13] CICECO-Aveiro Institute of Materials & Department of Medical Sciences, University of Aveiro, Aveiro, Portugal
[14] Department of Medical Sciences, University of Aveiro, Aveiro, Portugal
[15] Department of Biotechnology & Genetic Engineering, University of Development Alternative, Lalmatia, Dhaka, Bangladesh
[16] The Institute for Tropical Biology and Conservation, University Malaysia Sabah, Kota Kin-abalu, Sabah, Malaysia
[17] Hunter Medical Research Institute, New Lambton, Australia

Corresponding author
Veeranoot Nissapatorn, nissapat@gmail.com

## ABSTRACT

*Garcinia mangostana* L., also known as the mangosteen tree, is a native medicinal plant in Southeast Asia having a wide variety of pharmacologically active compounds, including xanthonoid mangostin. In this study, we examined the pharmacological activities of the selected semi-synthetic mangostin derivative, namely, amoebicidal activity, encystation inhibition, excystation activity, and removal capacity of adhesive *Acanthamoeba* from the surface of contact lens (CL).

Among the three derivatives, C1 exhibited promising anti-*Acanthamoeba* activity against *Acanthamoeba triangularis* WU19001 trophozoites and cysts. SEM images displayed morphological changes in *Acanthamoeba* trophozoites, including the loss of acanthopodia, pore formation in the cell membrane, and membrane damage. In addition, the treated cyst was shrunken and adopted an irregular flat cyst shape. Under a fluorescence microscope, acridine orange and propidium iodide (AO/PI) staining revealed C1 induced condensation of cytoplasm and chromatin with the loss of cell volume in the treated trophozoites, while calcofluor white staining demonstrated the leakage of cell wall in treated cysts, leading to cell death. Interestingly, at the concentration ranges in which C1 showed the anti-*Acanthamoeba* effects ($IC_{50}$ values ranging from 0.035–0.056 mg/mL), they were not toxic to Vero cells. C1 displayed the highest inhibitory effect on *A. triangularis* encystation at 1/16×MIC value (0.004 mg/mL). While C1 demonstrated the excystation activity at 1/128×MIC value with a high rate of 89.47%. Furthermore, C1 exhibited the removal capacity of adhesive *Acanthamoeba* from the surface of CL comparable with commercial multipurpose solutions (MPSs). Based on the results obtained, C1 may be a promising lead agent to develop a therapeutic for the treatment of *Acanthamoeba* infections and disinfectant solutions for CL.

activity, Contact lens, Semi-synthetic derivative

## INTRODUCTION

The number of contact lens (CL) wearers is continually increasing, and has been estimated at over 140 million CL users worldwide (*Harbiyeli et al., 2022*). Poor hygiene habits when handling CL, for example, inadequate lens cleaning, exposure of contact lenses to contaminated water, prolonged use, and sleeping with CL, can lead to an increased risk of developing infections keratitis (IK), which subsequently causes several clinical outcomes such as corneal melting and scarring, ocular morbidity, scleral inflammation, and blindness (*Cope et al., 2017*). A wide range of microorganisms can cause IK, for example, filamentous fungi (*Fusarium* spp. and *Aspergillus* spp.), bacteria (*Pseudomonas aeruginosa*, *Staphylococcus aureus*, *Streptococcus* spp., *Klebsiella* spp., *Citrobacter* spp., *Corynebacterium* spp., and *Propionibacterium* spp.), viruses, and protozoa (*Acanthamoeba* spp.) (*O'Callaghan et al., 2019*; *Lee et al., 2021*; *Ting et al., 2021*).

*Acanthamoeba* spp., opportunistic unicellular eukaryotic organisms, are free-living amoeba (FLA) that reside in a remarkably wide range of habitats, including soil, air, and water (*de Lacerda & Lira, 2021*). They can be found in various water sources, for example, pond water, swimming pools, beaches, tap water, and bottled water (*Esboei et al., 2020*; *de Lacerda & Lira, 2021*). *Acanthamoeba* spp. can cause rare and fatal brain infections, commonly known as granulomatous amoebic encephalitis (GAE) (*Sarink et al., 2022*). The mortality rate associated with GAE exceeds 90% and occurs mainly among individuals with metabolic, physiological, and immunological disorders or deficiencies (*Kalra et al.,*

*2020*). *Acanthamoeba* can also cause corneal infections and blindness, known as *Acanthamoeba* keratitis (AK), which is a sporadic disease often related to CL users, and the incidence is increasing in many countries, including the United States (*Maycock & Jayaswal, 2016*), New Zealand (*McKelvie et al., 2018*), and Denmark (*Nielsen, Ivarsen & Hjortdal, 2020*). *Acanthamoeba* has only two life cycle stages, transforming from an active form (trophozoite) to a dormant form (cyst) (*Fanselow et al., 2021*). The trophozoite is about 15–50 μm in size and appears irregular, oval, round, or pear-shaped with spine-like structures known as acanthopodia on its surface, which have been reported to show the ability to adhere to contact lenses (*de Lacerda & Lira, 2021*). *Acanthamoeba* transforms into a form of resistant cyst in harsh environments or conditions, such as starvation, exposure to biocides or extreme temperatures, pH, presence of chlorine, and changes in osmolarity (*Bergmanson et al., 2011*). The most routinely used anti-*Acanthamoeba* keratitis drugs are antiseptics chlorhexidine (CHX) and polyhexamethylene biguanide (PHMB). These drugs interact with the cytoplasmic membrane, causing the loss of cellular components. They also inhibit enzymes that are necessary for cell respiration (*Lorenzo-Morales, Khan & Walochnik, 2015*; *Fanselow et al., 2021*). However, the potential toxicity of PHMB on human corneal cells has been reported (*Lorenzo-Morales et al., 2013*). Therefore, the effort to develop safe and effective drugs remains a great challenge.

Various plants have been reported as an important source of amoebicidal agents against *Acanthamoeba* infection with high anti-amoebic activity and low toxicity (*Niyyati, Dodangeh & Lorenzo-Morales, 2016*). *Garcinia mangostana* L. (mangosteen, *Clusiaceae*) is a native of Southeast Asia. It has been used as a medicinal plant (*Nauman & Johnson, 2022*). Mangosteen pericarp (MP) is an important source of bioactive xanthones, which has remarkable antioxidant, anti-inflammatory, immunomodulatory, antiviral, antifungal, antibacterial, and anticancer activities (*Abate et al., 2022*; *Nauman & Johnson, 2022*). In literature, *G. mangostana* extract and α-mangostin have previously shown anti-*Acanthamoeba* activity and anti-adhesion properties against *Acanthamoeba triangularis* (*Sangkanu et al., 2021*; *Sangkanu et al., 2022*). In this study, mangostin isolated from the pericarps of *G. mangostana* was modified to three semi-synthetic derivatives using epoxidation and nitrile-addition reactions. The epoxy- and nitrile-mangostin derivatives have gained more interest due to the discovery of anti-parasitic activity against *Plasmodium falciparum* (*Mahabusarakam et al., 2006*) and antioxidant activity (*Mahabusarakam et al., 2009*). The pharmacological activities of these compounds named 1,3-dihydroxy-6-(2,3-epoxypropoxy)-7-methoxy-2,8-bis(3-methybut-2-enyl)-9H xanthen-9-one (C1), 1-Hydroxy-3,6-di(4-cyanopropoxy)-7-methoxy-2,8-bis(3-methylbut-2-enyl)-9H xanthen-9-one (C2), and 1,3-Dihydroxy-6-(4-cyanopropoxy)-7-methoxy-2,8-bis(3-methylbut-2-enyl)-9H-xanthen-9-one (C3) were investigated. To our knowledge, these semi-synthetic compounds have never been reported on anti-*Acanthamoeba* activities. This study, therefore, aims to (i) screen the anti-*Acanthamoeba* activity of mangostin derivatives, (ii) evaluate the inhibition of *Acanthamoeba* encystation and excystation activity of selected mangostin derivative, and (iii) estimate the effect of the selected compound on the adhesion of *Acanthmoeba* to the surface of CL. These studies

may provide more insights into the information of this selected compound and future therapeutic applications related to *Acanthamoeba* infection or CL wearer.

# MATERIALS AND METHODS

## Chemicals used

Proteose peptone, Page's saline, and yeast extract were procured from HiMedia Laboratories (Mumbai, India). Sodium citrate dihydrate ($C_6H_5Na_3O_7.2H_2O$), disodium phosphate ($NaHPO_4$), sodium chloride ($NaCl$), calcium chloride ($CaCl_2$), glucose, and phenylmethylsulphonyl fluoride (PMSF) were from Sigma Chemical Co. (St. Louis, MO, USA). Potassium dihydrogen phosphate ($KH_2PO_4$) and magnesium sulfate heptahydrate ($MgSO_4.7H_2O$) were procured from Labscan (Bangkok, Thailand). Trypan blue (0.4%) was obtained from Gibco BRL (Grand Island, NY, USA). All chemicals and medium components used were of analytical grade.

## Preparation of mangostin and its derivatives

The extraction and isolation procedures of mangostin were previously described by *Mahabusarakam et al. (2009)*. Briefly, 4 kg of air-dried powdered pericarps of *G. mangostana* (*Mahabusarakam et al., 2009*) were soaked in 2.0 L of dichloromethane ($CH_2Cl_2$) for 48 h. The extraction was repeated twice, and the combined solution was filtered through Whatman No. 1 filter paper (GE Healthcare Life Science, Buckinghamshire HP7 9NA, United Kingdom) using a vacuum and pressure pump. The collected filtrate was evaporated to dryness under a vacuum at 45 °C using a rotary evaporator (Hei-VAP Advantage HL/G3; Heidolph, Schwabach, Germany). The yellow precipitate was collected. The solid was re-dissolved in dichloromethane and partitioned with 20% (w/v) sodium carbonate solution. The organic layer was separated and washed with water several times until the water layer was no longer basic. The dried organic layer was concentrated, and mangostin was obtained as a yellow crystalline solid.

To prepare epoxide derivative, a mixture of mangostin (5 g, 12 mmol) in dimethylformamide (DMF) (20 mL) and sodium hybrid (2 g) in DMF (10 mL) was stirred at room temperature for 30 min, followed by the addition of 2.5 mL 1-chloro-2,3-epoxypapane. The reaction mixture was then refluxed for 6 h and worked up. The residue was purified by silica gel chromatography and eluted with light petroleum/dichloromethane (4:5) to obtain 1,3-Dihydroxy-6-(2,3-epoxypropoxy)-7-methoxy-2,8-bis(3-methylbut-2-enyl)-9H-xanthen-9-one (C1) (1.5 g, 26% yield) (*Mahabusarakam et al., 2006*).

The preparation of nitrile derivatives was previously described by *Mahabusarakam et al. (2009)*. A solution of mangostin (2 g, 4.8 mmol) in DMF (10 mL) and sodium hydride (2 g) in DMF (5 mL) was stirred at room temperature for 30 min, followed by addition of 4-chlorobutyronitrile (1.5 mL, 15 mmol). The reaction mixture was then refluxed for 5 h, poured into iced water, neutralized with 3 M hydrochloric acid (HCl), and extracted with dichloromethane. Later, the dichloromethane layer was washed with water and dried over sodium sulfate. The solvent was removed, and the residue was purified by silica gel chromatography. The sample was eluted with the mixed solvent of hexane/dichloromethane

(1:8) to give yellow needles of 1-Hydroxy-3,6-di(4-cyanopropoxy)-7-methoxy-2,8-bis (3-methylbut-2-enyl)-9H-xanthen-9-one (C2) (1 g, 43% yield) and 1,3-Dihydroxy-6-(4-cyanopropoxy)-7-methoxy-2,8-bis(3-methylbut-2-enyl)-9H-xanthen-9-one (C3) (0.25 g, 9% yield).

## Strain and growth conditions

*Acanthamoeba triangularis* WU19001 genotype T4 (MW647650) used in this study was previously isolated from the recreational reservoir of Walailak University, Nakhon Si Thammarat, Thailand, by *Mitsuwan et al. (2020)*. The strain was seeded in T-25 tissue culture flasks (SPL Life Science Co., Ltd., Gyeonggi-do, South Korea) with 10 mL of peptone–yeast extract–glucose (PYG) medium containing (g/L) proteose peptone 20, yeast extract 2.0 and glucose 18, $C_6H_5Na_3O_7.2H_2O$ 1.0, $CaCl_2$ 0.059, $MgSO_4.7H_2O$ 0.98, sodium phosphate dibasic heptahydrate ($Na_2HPO_4.7H_2O$) 0.355, $KH_2PO_4$ 0.34, ammonium iron (II) sulfate hexahydrate [$Fe(NH_4)_2(SO_4)_2.6H_2O$] 0.02 in 1,000 mL distilled water pH 7.0. After incubation in a dark room at ambient temperature for 72 h, *A. triangularis* trophozoites were observed.

For *A. triangularis* cyst preparation, the trophozoites grown in PYG medium for 72 h were centrifuged at 4,000 rpm for 5 min in a Sorvall ST 40R centrifuge (Thermo Scientific, Waltham, MA, USA) and then washed twice with Neff's encystment medium pH 8.9 (*Neff et al., 1964*) contained (g/L) KCl 7.455, ammediol (2-amino-2-methyl-1,3-propanediol) 2.44, $NaHCO_3$ 0.084, $MgSO_4.7H_2O$ 1.968, $CaCl_2.2H_2O$ 0.0588. The trophozoites were transferred into T-25 tissue culture flasks containing 10 mL of an encystment medium, and encystation was induced in a dark room at ambient temperature for 7 days. In order to obtain only mature cysts, the cells were collected and then centrifuged at 4,000 rpm for 5 min and treated with sodium dodecyl sulfate (SDS) solution at 0.5% (w/v) final concentration for 20 min to solubilize trophozoites and immature cysts while mature cysts were resistant to SDS solution (*Moon et al., 2014*). Later, the mature cysts were harvested by centrifugation at 4,000 rpm for 5 min and washed twice with 0.85% (w/v) NaCl solution. The obtained cysts were re-suspended in Neff's encystment medium.

The viability of *A. triangularis* trophozoites and cysts was determined by trypan blue exclusion staining. *A. triangularis* suspension was mixed with the same volume of 0.2% trypan blue and incubated at ambient temperature for 3 min (*Sangkanu et al., 2022*). The viable (unstained) and dead (stained dark blue) amoebae were enumerated using a hemocytometer (Tiefe Depth Profondeur, Marienfeld, Germany) under an inverted microscope (Nikon ECLIPSE TE2000-S, Tokyo, Japan).

## Screening of anti-*Acanthamoeba* activity of mangostin derivatives

Each of the compounds was screened for anti-*Acanthamoeba* activity against *A. triangularis* trophozoites and cysts. Briefly, each of these compounds was dissolved in 100% dimethyl sulfoxide (DMSO) to a 100 mg/mL final concentration and stored at −20 °C until use. The stock solution of each compound was diluted with the cultivation medium to give the final concentration of 2 mg/mL, and the 50 µL of aliquots were added to a 96-well plate (SPL Life Sciences, Seoul, South Korea). *A. triangularis* trophozoites and

cysts were prepared as described above, and an aliquot of 50 µL of cell suspension prepared at $2 \times 10^5$ cells/mL was inoculated into each well of the plate, followed by incubation in a dark room at ambient temperature for 24 h. The final concentration of DMSO was 1% (v/v), and medium containing 1% (v/v) DMSO was used as a negative control, while a final concentration of 0.016 and 0.064 mg/mL chlorhexidine (CHX) solution was used as a positive control for trophozoite and cyst, respectively. The trypan blue exclusion staining was used to determine the number of viable cells present in samples as described above.

## Determination of minimal inhibitory concentration (MIC) of mangostin derivatives

The microtiter broth dilution method performed using a 96-well plate was used to determine the MIC value of semi-synthesized compounds against *A. triangularis* trophozoites and cysts according to the method of *Mitsuwan et al. (2020)*. In this assay, each of the compounds was diluted to give a final concentration of 0.016–1024 mg/mL. The cell suspension was prepared as described above, after which 50 µL of an aliquot of the cell suspension ($2 \times 10^5$ cells/mL) was seeded into each well and incubated at ambient temperature under darkness for 24 h. *A. triangularis* trophozoites and cysts exposed to 1% (v/v) DMSO and CHX solutions (a final concentration of 0.008–0.128 mg/mL) were used as a negative and positive control, respectively. The plates were incubated in a dark room at ambient temperature for 24 h. The viability was calculated according to the following equation:

% Viability = (Mean of the viable parasite/Negative control) × 100

MIC is defined as the lowest concentration of the tested compound that inhibits visible growth of at least 90% of *A. triangularis* (90% of amoeba died).

## Determination of inhibitory concentration 50 (IC$_{50}$)

IC$_{50}$ of C1 and CHX against *Acanthamoeba* trophozoites and cysts was further determined. The protocol was modified from *Kolören et al. (2019)*. In brief, the assay was performed in a 96-well plate. C1 and CHX were prepared with two-fold serial dilution (a final concentration of 0.015–1 mg/mL). Then, 50 µL of $2 \times 10^5$ cells/mL of trophozoites and cysts were added to each well and incubated in a dark room at ambient temperature for 24 h. The untreated parasite [medium containing 1% (v/v) DMSO] was included as a negative control. After incubation for 24 h, trypan blue staining was used to determine the number of viable trophozoites and cysts by counting on a hemocytometer. The results are given as percent inhibition compared to control cells (considered as 100%). The IC$_{50}$ was calculated using the GraphPad Prism, version 7 for Windows (GraphPad Software, San Diego, CA, USA). This experiment was performed in triplicate.

## Scanning electron microscopy (SEM) analysis

*A. triangularis* trophozoites and cysts treated with C1 were carried out to observe any morphological changes using SEM (Zeiss, Munich, Germany) at the Center for Scientific and Technological Equipment, Walailak University, Nakhon Si Thammarat, Thailand

(*Mitsuwan et al., 2020*). The trophozoites were treated with 0.064 mg/mL C1 (1/2×MIC) and CHX 0.008 mg/mL (1/2×MIC). *A. triangularis* cysts were also treated with CHX (0.032 mg/mL) and C1 (0.032 mg/mL) at 1/2×MIC of CHX. After incubation in a dark room at ambient temperature for 24 h, cells were collected by centrifugation at 4,000 rpm for 5 min and washed three times in 0.01 M phosphate-buffered saline (PBS) solution pH 7.4. Cells were then smeared very thinly onto a sterile cover glass and air-dried. The dried cover glass was placed in a 24-well plate and fixed with 2.5% glutaraldehyde solution overnight at ambient temperature. The samples were then washed three times with PBS solution and dehydrated in a graded series of ethanol solutions from low- to high-density (20%, 40%, 60%, 80%, 90%, and 100%) for 30 min at each step, further mounted on aluminum stubs, and allowed to dry using a critical point dryer. The morphology of *A. trianguaris* trophozoites and cysts was subsequently analyzed under SEM.

## Fluorescence microscopy by acridine orange and propidium iodide staining

*A. triangularis* cell death after exposure to C1 was identified by AO/PI (acridine orange/ propidium iodide) double-staining assay according to the modified method of *Kusrini et al. (2020)*. For the staining, *A. triangularis* trophozoites were treated with CHX (0.008 mg/mL) and C1 (0.064 mg/mL) at 1/2×MIC in 1.5 mL microcentrifuge tubes, followed by mixing. After incubation for 24 h, samples were centrifuged at 4,000 rpm for 5 min, and the supernatant was discarded and the pellet was washed twice with 0.01 M PBS solution pH 7.4. The pellets were re-centrifuged at 4,000 rpm for 5 min and re-suspended in 100 μL AO/PI staining solution, which was prepared by adding 200 μL of AO (1 mg/mL) and 200 μL PI (1 mg/mL) (Sigma Chemical Co., St. Louis, MO, USA) in 600 μL 0.01 M PBS solution pH 7.4. The cell suspension was then incubated in a dark room at ambient temperature for 10 min since both dyes are light-sensitive and placed onto a slide, carefully covered with a coverslip (*Momčilović et al., 2019*). The cells were visualized using a fluorescent microscope (Leica TCS SP5 laser scanning confocal microscope; Leica microsystem, Inc., Wetzlar, Germany). Live cells turned into green (AO), whereas the dead appeared red (PI).

## Calcofluor white staining

*A. triangularis* cysts were incubated in the absence or presence of C1 at 1/2×MIC (0.032 mg/mL) for 24 h. The cysts were harvested by centrifugation at 4,000 rpm for 5 min and washed twice with 0.01 M PBS solution pH 7.4. *A. triangularis* cysts were re-suspended in 2.5% calcofluor white (CFW) staining solution (Sigma Chemical Co., St. Louis, MO, USA) and incubated in a dark room at ambient temperature for 120 min and collected by centrifugation at 4,000 rpm for 5 min. Later, it was washed twice with 0.01 M PBS solution pH 7.4 to remove excess CFW stain and re-suspended in the same buffer solution. Finally, 20 μL cyst suspensions were placed onto a slide, carefully covered with a coverslip, and examined under Olympus, BX-53 fluorescent microscope (Olympus, Tokyo, Japan) with the excitation of 405 nm and emission band pass 420–480 nm.

## Cytotoxicity assay

The *in vitro* cytotoxicity of C1 and CHX was evaluated against the Vero cell (Elabscience, Wuhan, Hubei, China) according to the method of *Eawsakul et al. (2017)*. Vero cells were maintained in Dulbecco's Modifed Eagle's (DMEM) medium (Merck KGaA, Darmstadt, Germany) supplemented with 10% (v/v) fetal bovine serum (FBS) and 1% antibiotic cocktail containing penicillin G of 100 units/mL and streptomycin of 100 μg/mL. The culture was incubated at 37 °C in a humidified 5% (v/v) $CO_2$ 95% air atmosphere. The medium was changed every 2 days until 90% confluence was achieved. The cells were then enzymatically detached from the T-flask by adding trypsin- ethylenediaminetetraacetic acid (EDTA) solution (0.25% trypsin, 1 mM EDTA), and subsequently incubated at 37 °C in a humidified incubator (Binder, Tuttlingen, Germany) containing 95% room air/5% $CO_2$. After incubation, 100 μL of the suspensions were seeded in 96-well plates at a density of $1 \times 10^5$ cells/well, and then 100 μL of C1 and CHX at different concentrations was added. Positive and negative controls were prepared by adding the same amount of 1% (v/v) Triton X-100 and 0.01 M PBS solution pH 7.4, respectively. After incubation for 24 h, the cell viability of treated Vero cells was determined using a 3-(4,5-dimethylthiazol-2-yl)-2,5-diphenyltetrazolium bromide (MTT) assay. The medium was removed and cells were incubated with fresh medium containing 0.5 mg/mL of MTT for 4 h. The formed formazan crystal was dissolved by adding 100 mL of DMSO. The absorbance was measured using a microplate reader (Metertech M965; Metertech, Taipei, Taiwan) at 570 nm. The cell viability was calculated using the following equation. Based on ISO 10993-5 standard, the percentage of cell viability above 80% are non-cytotoxic; within 80–60% weak; 60–40% moderate, and below 40% strong cytotoxic, respectively (*International Organization for Standardization ISO 10993-5, 2009*).

Cell viability (%) = (ABt/ABu) × 100

ABt and ABu are the absorbance values of treated and untreated cells, respectively.

## Effect of mangostin derivative on *A. triangularis* encystation

The effect of C1 on *A. triangularis* encystation was assessed in 96-well plates according to the modified method of *Fakae et al. (2020)*. In brief, C1 was diluted with Neff's encystment medium to give a final concentration of 1/2×MIC–1/32×MIC. Then, 50 μL of each dilution was added to each well, and an aliquot of 50 μL containing approximately $6 \times 10^5$ trophozoites/mL was seeded into each well of the 96-well plate. The trophozoites in the presence of 10 mM phenylmethylsulfonyl fluoride (PMSF) were used as a positive control, whereas the trophozoites exposed to an encystment medium containing 1% (v/v) DMSO served as a negative control. The plates were incubated in a dark room at ambient temperature for 72 h. The viability of *A. triangularis* cells was quantified by trypan blue exclusion staining using a hemocytometer. To solubilize all trophozoites, a final concentration of 0.5% (w/v) SDS solution was added and incubated at ambient temperature for 30 min with mild shaking, and cysts were enumerated using a

hemocytometer. The inhibition effect of mangostin derivative on encystation was calculated according to the following equation:

$$\text{Percentage of encystation inhibition} = 100 - (\text{Post-digestion number}/\text{Pre-digestion number}) \times 100$$

## Effect of mangostin derivative on excystation activity of *A. triangularis*

The excystation assay was performed to determine the effect of C1 on excystation activity of *A. triangularis*. The protocol for this assay was obtained from *Anwar et al. (2019)* with a slight modification. C1 was diluted with a PYG growth medium to give a final concentration of 1×MIC–1/128×MIC. An aliquot of 50 µL of each dilution was added to each well, and cell suspension containing $6 \times 10^5$ cysts/mL was prepared as described above and followed by adding 50 µL of this into each well. The cysts in the presence of 10 mM PMSF were served as a negative control, while the cysts without C1 exposure served as a positive control. After incubation in a dark room at ambient temperature for 7 days (*Shing, Balen & Debnath, 2021*), a final concentration of 0.5% (w/v) SDS solution was added to solubilize all trophozoites and incubated at ambient temperature for 30 min with mild shaking, and cysts were enumerated using a hemocytometer. The excystation rate was calculated according to the following equation:

$$\text{Excystation rate (\%)} = (\text{No. of total cells} - \text{No. of cyst}/\text{No. of total cells}) \times 100$$

The results are representatives of three experiments presented as mean ± standard error.

## Effect of mangostin derivative on the adhesion of *A. triangularis* to the surface of contact lenses (CL)

The potential of C1 in removing *A. triangularis* trophozoites and cysts adherent to contact lens (CL) was evaluated according to a previous study (*Mitsuwan et al., 2020*) with slight modification. In this assay, the commercial CL-1 was removed from the original package and placed in a 24-well plate. Each lens was rinsed twice with 0.01 M PBS solution pH 7.4. *A. triangularis* trophozoites and cysts were prepared as described above, and an aliquot of 500 µL cell suspension containing $1 \times 10^6$ cells/mL was dropped on the CL in a 24-well plate and incubated in a dark room at ambient temperature for 24 h. C1 and CHX at a final concentration of 1/2×MIC (500 µL) were added to each well and incubated at ambient temperature under darkness for 24 h. Two brands of multi-purpose disinfecting (MPD) solutions, including Renu®freshTM (Bausch & Lomb, Rochester, NY, USA) (MPD-1) and Duna NO RUB (Alcon Laboratories, INC, Fort Worth, TX, USA) (MPD-2) were used as positive controls. Contact lenses (CL) were gently washed with 0.01 M PBS solution pH 7.4 and transferred to a microcentrifuge tube. The samples were then placed in an ice bath for 10 min and shaken in a vortex for 5 min. The total number of cells in the microcentrifuge tube was counted under an inverted microscope.

**Figure 1  The chemical structures of semi-synthetic mangostin derivatives.** C1 named 1,3-Dihydroxy-6-(2,3-epoxypropoxy)-7-methoxy-2,8-bis(3-methylbut-2-enyl)-9H-xanthen-9-one, C2 named 1-Hydroxy-3,6-di(4-cyanopropoxy)-7-methoxy-2,8-bis(3-methylbut-2-enyl)-9H xanthen-9-one, and C3 named 1,3-Dihydroxy-6-(4-cyanopropoxy)-7-methoxy-2,8-bis(3-methylbut-2-enyl)-9H-xanthen-9-one.

## Statistical analysis

All experiments were carried out in triplicate, and the mean values were registered. The data were compared by one-way ANOVA and the Tukey's t-test using the statistical package software (SPSS version 26.0, Chicago, IL, USA). Data were expressed as mean ± SD. In all analyzes, $p < 0.05$ was considered statistically significant.

## RESULTS

### Effect of mangostin derivatives against *A. triangularis*

The anti-*Acanthamoeba* activity of the mangostin derivatives at a final concentration of 1.0 mg/mL against *A. triangularis* WU19001 trophozoites and cysts was screened through the microtiter broth dilution method. Three semi-synthetic mangostin derivatives were obtained: 1,3-Dihydroxy-6-(2,3-epoxypropoxy)-7-methoxy-2,8-bis(3-methylbut-2-enyl)-9H-xanthen-9-one (C1, epoxide derivative), 1-Hydroxy-3,6-di(4-cyanopropoxy)-7-methoxy-2,8-bis(3-methylbut-2-enyl)-9H xanthen-9-one (C2, nitrile derivative) and 1,3-Dihydroxy-6-(4-cyanopropoxy)-7-methoxy-2,8-bis(3-methylbut-2-enyl)-9H-xanthen-9-one (C3, nitrile derivative), and their chemical structures were shown in Fig. 1. These tested compounds exhibited inhibitory activities at 1 mg/mL and were further evaluated for their MICs. Among these, C1 showed greater anti-*Acanthamoeba* activity at 0.128 and 0.064 mg/mL on trophozoites and cysts, respectively (Table 1). Moreover, C1 exhibited anti-*Acanthamoeba* cysts activity similar as chlorhexidine (CHX). Regarding to mangostin nitrile derivatives included C2 and C3 on *A. triangularis* trophozoites and cysts, both exhibited much higher MIC values *in vitro* against trophozoites (0.512 mg/mL) and cyst

**Table 1 Screening of anti-*Acanthamoeba* activity and minimal inhibitory concentration (MIC) of semi-synthetic mangostin derivatives and chlorhexidine against *A. triangularis* WU19001 trophozoites and cysts.**

| Anti-*Acanthamoeba* agent | Anti-*Acanthamoeba* activity at a final concentration of 1 mg/mL | | MIC (mg/mL) | |
| --- | --- | --- | --- | --- |
| | Trophozoite | Cyst | Trophozoite | Cyst |
| C1 | + | + | 0.128 | 0.064 |
| C2 | + | + | 0.512 | 1.024 |
| C3 | + | + | 0.512 | 1.024 |
| Chlorhexidine | + | + | 0.016 | 0.064 |

Note:
Symbol: +, positive with inhibition of more than 90% of viable growth compared with the control.

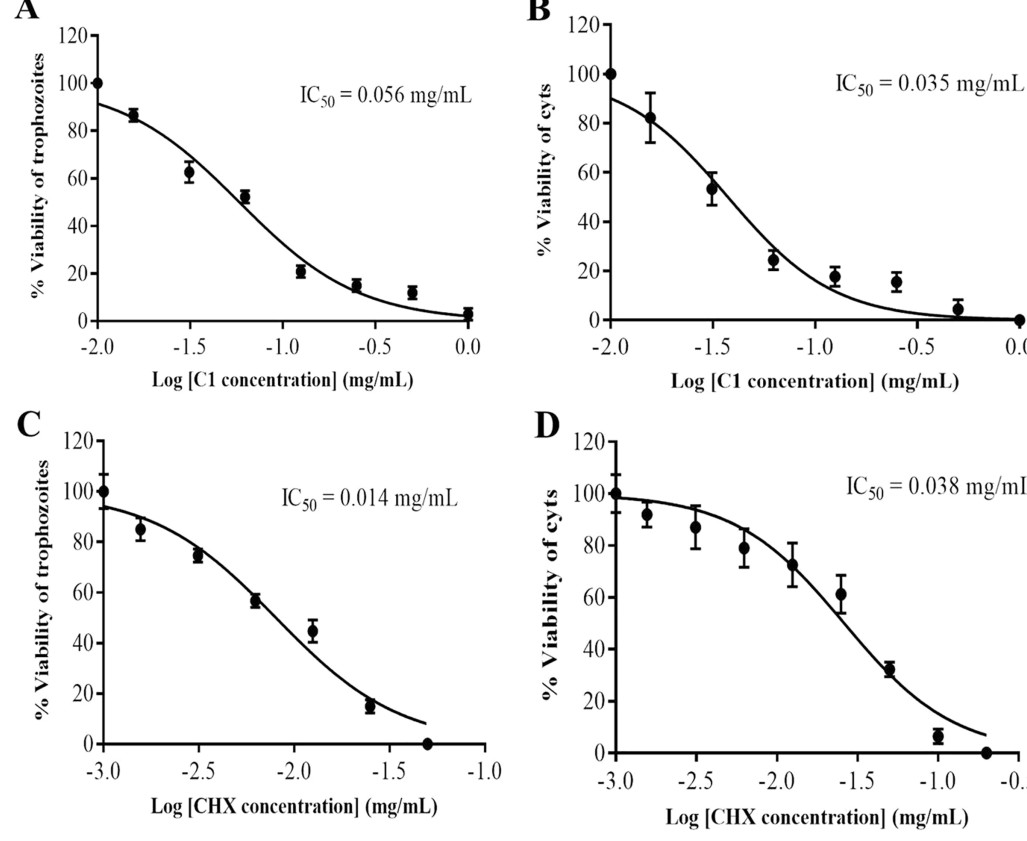

**Figure 2 Identification of mangostin derivative (C1) $IC_{50}$ and chlorhexidine (CHX) $IC_{50}$.** (A–B) Representative images of C1 $IC_{50}$ towards trophozoites and cysts, respectively. (C–D) Representative images of CHX $IC_{50}$ towards trophozoites and cysts, respectively. Values are presented as mean ± standard deviation (SD) of triplicate determinations ($n = 3$).

(>1 mg/mL) compared to those of CHX. C1 and CHX also were shown in demonstrating the inhibitory activity against *Acanthamoeba* trophozoites with $IC_{50}$ of 0.056 and 0.014 mg/mL, respectively (Fig. 2). In addition, C1 and CHX exhibited the inhibitory effect
C1  Chlorhexidine  Untreated

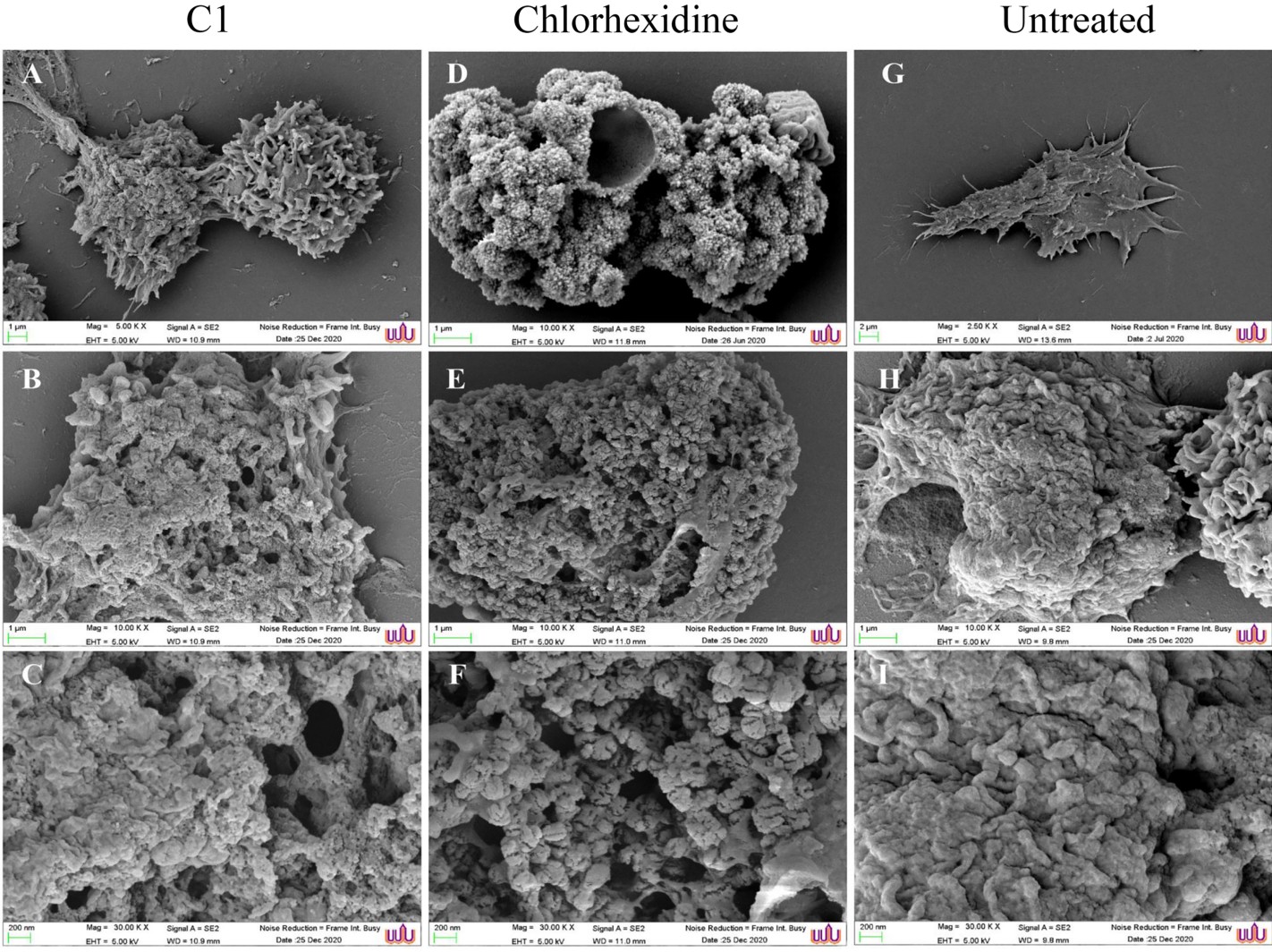

**Figure 3 Morphological changes of *Acanthamoeba triangularis* trophozoites after exposure to mangostin derivative (C1).** (A–C) Treated cell with 1/2 MIC C1. (D–F) Treated cells with 1/2×MIC chlorhexidine (CHX) and (G–I) treated cells with 1% (v/v) DMSO. CHX and DMSO were used as positive and negative controls, respectively. Morphological changes of the *A. triangularis* were observed by SEM. Magnifications were revealed as: A, D, G = 5,000×; D, E, F = 10,000×; G, H, I = 30,000×.

against cysts with IC$_{50}$ values of 0.035 and 0.038 mg/mL, respectively (Fig. 2). In this regard, C1 was selected for further study.

## Effect of mangostin derivative on morphological changes of *A. triangularis*

The morphological changes of the *A. triangularis* trophozoites upon exposure to C1 at its 1/2×MIC value (0.064 mg/mL) were observed by scanning electron microscopy (SEM). The *A. triangularis* trophozoites exhibited spine-like structures on their surface known as acanthopodia. Based on the SEM images, shortening and loss of acanthopodia on the amoeba membrane were observed after treated trophozoites with C1 (0.064 mg/mL) (Fig. 3A) and CHX (Fig. 3D) for 24 h. Moreover, trophozoites treated with CHX at

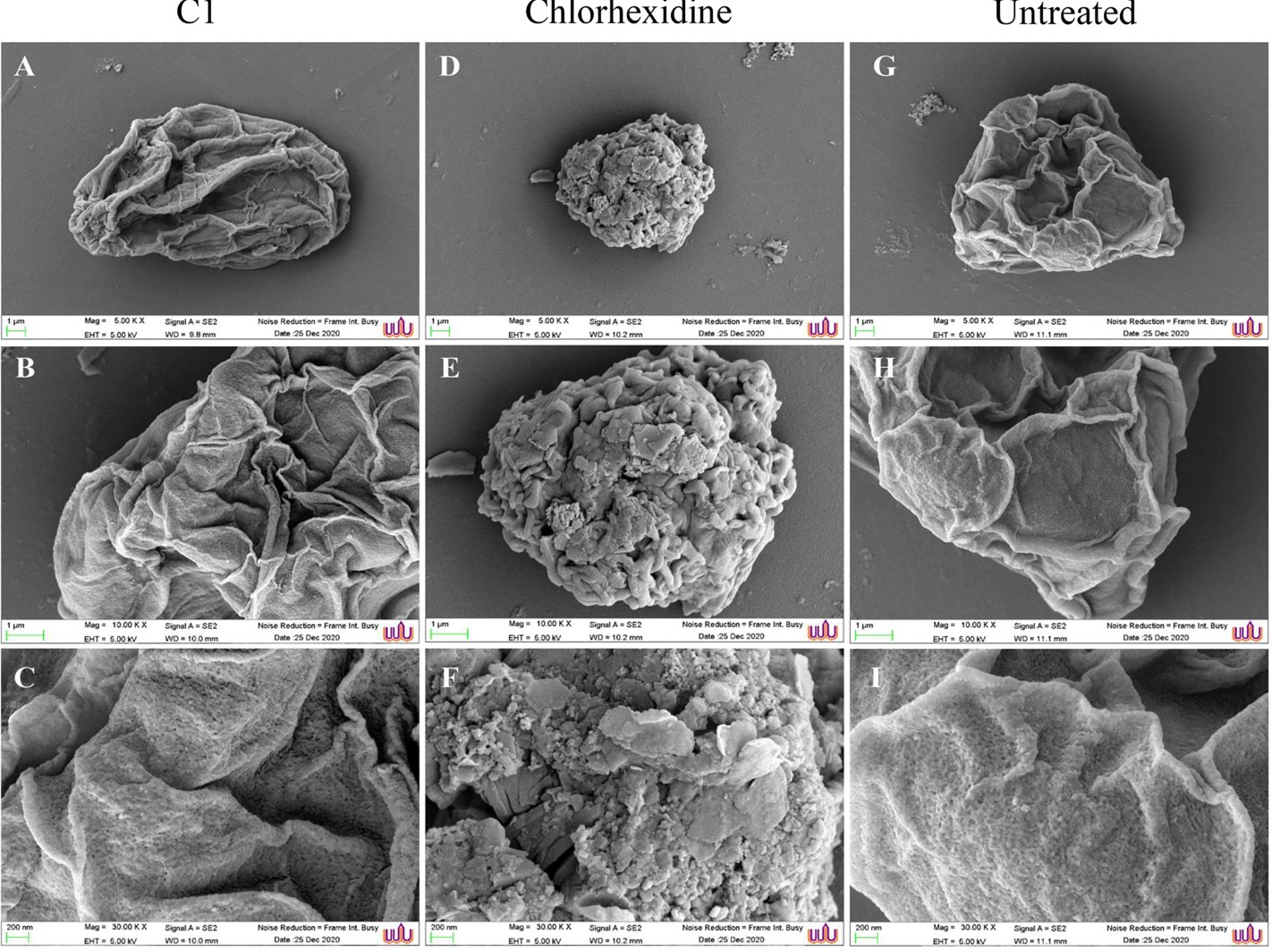

**Figure 4 Morphological changes of *Acanthamoeba triangularis* cysts after exposure to mangostin derivative (C1).** (A–C) Treated cell with 1/2×MIC C1. (D–F) Treated cell with 1/2×MIC chlorhexidine (CHX) and (G–I) treated cell with 1% (v/v) DMSO. CHX and DMSO were used as positive and negative controls, respectively. Morphological changes of *A. triangularis* were observed by SEM. Magnifications were revealed as: A, D, G = 5,000×; D, E, F = 10,000×; G, H, I = 30,000×.

1/2×MIC value (0.008 mg/mL) showed stronger loss of acanthopodia compared to that of C1. On the contrary, untreated cells exhibited numerous acanthopodia on the amoebic surface (Fig. 3G). The most remarkable changes on the amoeba morphology after treated with C1 and CHX were the reduction of cell size, development of abnormal shapes, blebbing on the plasma, pore formation in the cell membrane, and membrane damage (Fig. 3). A large number of pores were detected in treated trophozoites with CHX when compared to those of C1.

SEM images displayed morphological changes in *A. triangularis* cysts treated with C1 (1/2×MIC value = 0.032 mg/mL). In this condition, cysts became shrunk and showed thick ridges on the wrinkled surface with the observation of flat and irregular shapes (Figs. 4A–4C). The normal morphological characteristics such as triangular shape, smooth

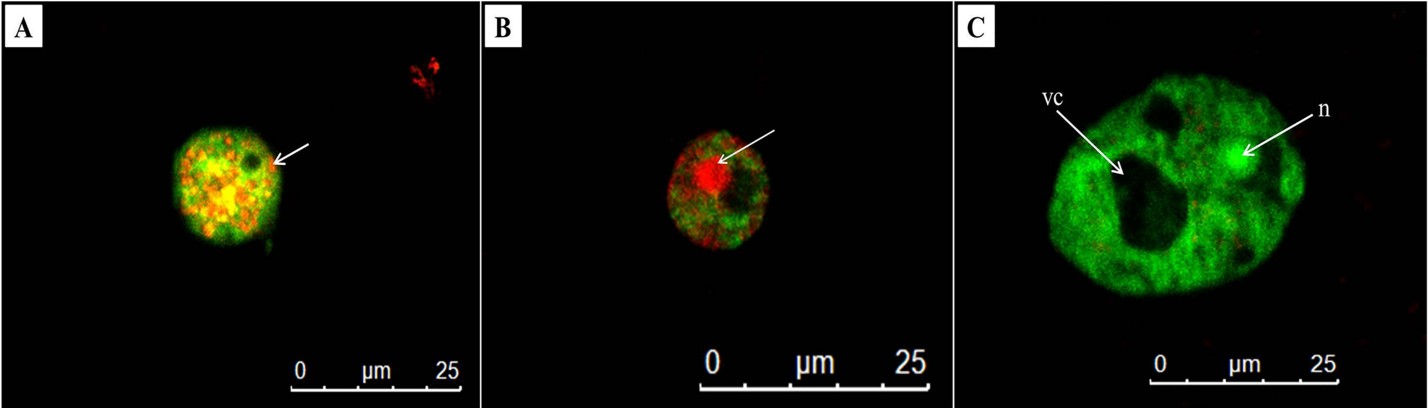

**Figure 5 Representative images of *Acanthamoeba triangularis* trophozoites stained with AO/PI dye under fluorescent microscopy.** (A) Treat cells with 1/2×MIC value of mangostin derivative (C1). (B) Treated cells with 1/2×MIC value of chlorhexidine (CHX). (C) Untreated cells.

surface, and appearance of thin ridges on the wrinkled surface were observed in the untreated cysts (Figs. 4G–4I). The cyst size of treated ones with CHX at 1/2×MIC value (0.032 mg/mL) was smaller compared with treated cysts with C1 (Figs. 4A and 4D). In addition, the result also showed that the cysts treated with CHX had rougher surfaces compared to those treated with C1.

## Determination of cell death by fluorescence microscopy-acridine orange/propidium iodide (AOPI) staining

AO/PI double staining is a technique used to differentiate between viable and non-viable cells. In untreated trophozoites, the cells were observed the green cytoplasm and bright green fluorescent of an intact nucleus stained by AO, indicating healthy and viable *Acanthamoeba* cells (Fig. 5C). In addition, a prominent vacuole was seen in untreated *Acanthamoeba* trophozoite. Meanwhile, *A. triangularis* cells treated with C1 and CHX were observed as orange to red granules in cells, indicating non-viable *Acanthamoeba* cells (Figs. 5A and 5B).

## Detection of cysticidal activity by calcofluor white staining

The cysticidal activity of C1 on *A. triangularis* cysts was evaluated by calcofluor white (CFW) staining assay. This fluorescence stain is known to bind to the cellulose deposition in the cell wall layer of *Acanthamoeba* cysts (*El-Sayed & Hikal, 2015*). The cell walls of *Acanthamoeba* cyst consisting of two layers identified as ectocyst and endocyst were observed in untreated cysts (Figs. 6A, 6B and 6E). Furthermore, our results showed that C1 and CHX had cysticidal activities and caused cyst death based on the loss of CFW staining (Figs. 6C and 6D).

## *In vitro* cell cytotoxicity

The cytotoxicity of C1 and CHX towards Vero cells was investigated using the MTT assay. The $IC_{50}$ values of C1 for *Acanthamoeba* trophozoites and cysts were 0.056 and 0.035 mg/mL, respectively. According to ISO 10993-5, at the cell viability exceeds 80%,
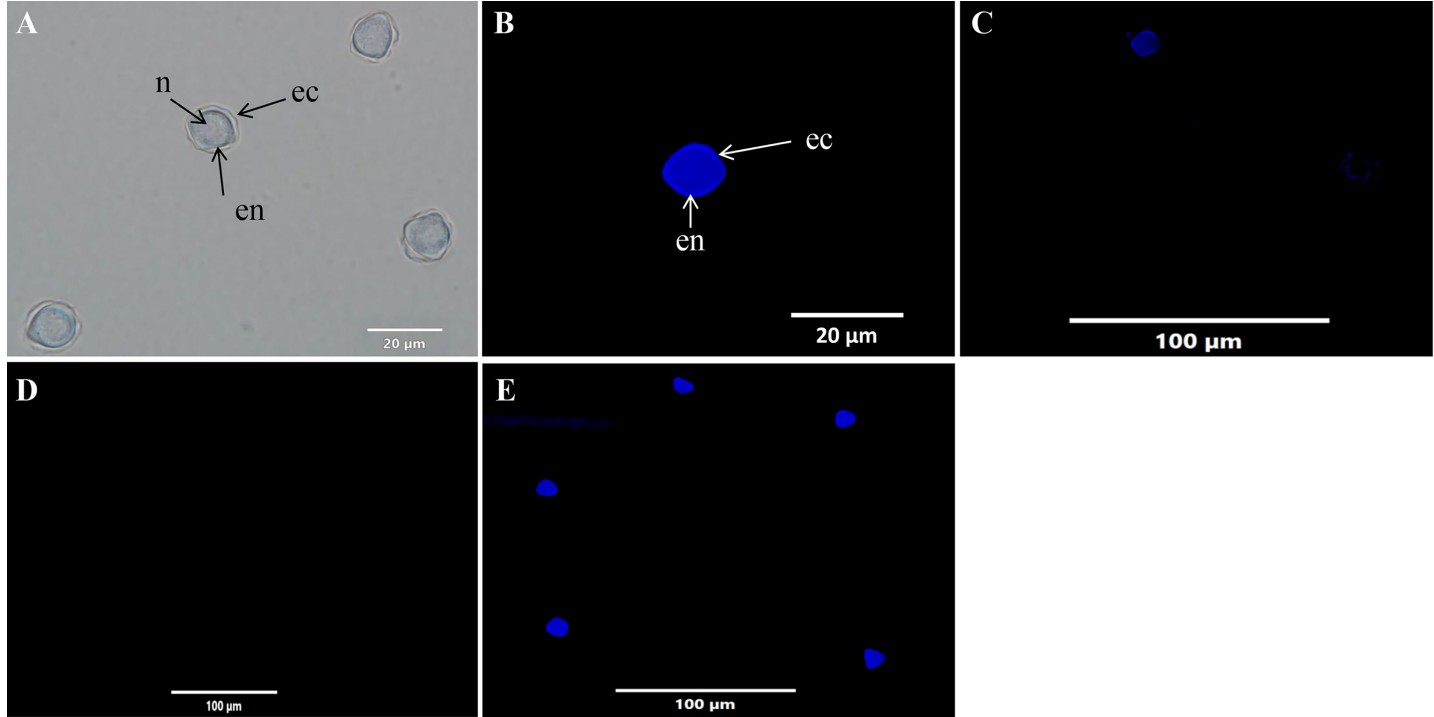

**Figure 6** *Acanthamoeba triangularis* **cysts stained with calcofluor white (CFW) dye under fluorescent microscopy.** (A) Cyst-specific staining with CFW dye under light microscope. (B) Healthy cyst-specific staining using CFW dye under fluorescent microscope. (C) Treated cysts with 1/2×MIC value of mangostin derivative (C1) and stained with CFW dye under fluorescent microscope. (D) Treated cells with 1/2×MIC value of chlorhexidine (CHX) and stained with CFW dye under fluorescent microscope. (E) Untreated cysts and stained with CFW dye under fluorescent microscope. Magnifications were revealed as: A, B = 1,000×; C, D, E = 400×.

compound is considered non-cytotoxic, while at cell viability range between 60–80%, as weakly cytotoxic. In this study, the non-cytotoxic dose of C1 was 0.04 mg/mL and the weakly cytotoxic concentration of C1 was 0.08 mg/mL (Fig. 7). Therefore, the results confirmed that C1 had no cytotoxic effect on Vero cells at inhibitory concentrations. The $IC_{50}$ values of CHX for *Acanthamoeba* trophozoites and cysts were 0.014 and 0.038 mg/mL, respectively and this obtained data indicated that CHX had strong cytotoxic effect on Vero cells.

## Effect of mangostin derivative as an inhibitor on *A. triangularis* encystation

Encystation of *Acanthamoeba* leads to cyst formation from vegetative trophozoites. This is essential for parasite survival under harsh conditions. The inhibitory effect of C1 on *A. triangularis* encystation was evaluated. Based on the results obtained, C1 displayed a high inhibitory effect on *A. triangularis* encystation at 1/2×MIC (0.064 mg/mL) to 1×32 MIC (0.008 mg/mL) with encystation inhibition of 6.0–9.0% (Fig. 8). Of this, the best encystation inhibition was observed at 1/16×MIC (0.004 mg/mL) (Fig. 8). Meanwhile, 10 mM PMSF was used as a positive control and suppressed up to 90% of trophozoites from forming into cysts (Fig. 8). C1 gave a better inhibitory effect against *A. triangularis* encystation compared to control group (without C1) (Fig. 8).

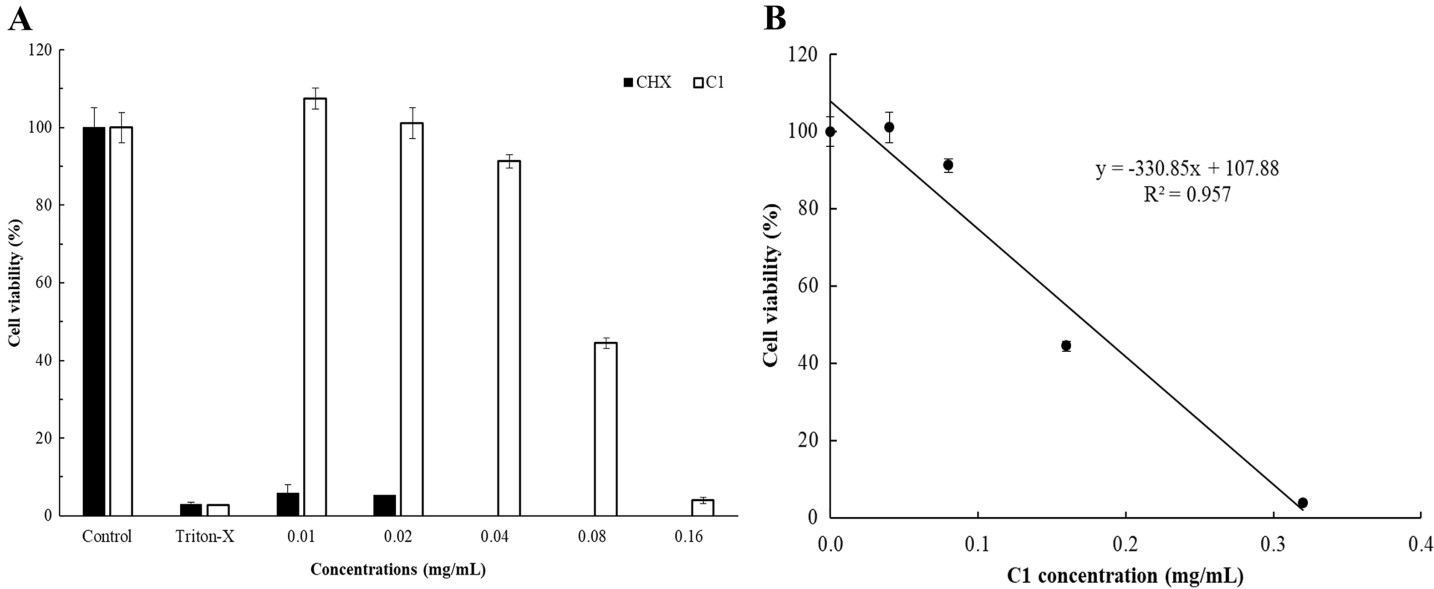

**Figure 7** *In vitro* **cytotoxic effect of mangostin derivative (C1) and chlorhexidine (CHX) against Vero cells.** The cytotoxic activity of C1 (A and B) and CHX (A) on Vero cells was evaluated after incubation at 37 °C for 24 h.     

## Effect of mangostin derivative on the excystation activity of *A. triangularis*

Excystation assay was conducted to determine the effect of C1 on the ability of *A. triangularis* to re-emerge from the cyst stage to trophozoites. Cysts treated with PMSF (10 mM) were used as a negative control, while the cysts without C1 exposure were used as a positive control. According to the results, the treatment of amoebic cysts with C1 significantly succeeded in triggering excystation ($p < 0.05$) at 1/128×MIC, as compared to the control groups (10 mM PMSF and without adding C1) (Fig. 9) with excystation rate of 89.47%. In addition, treatment of *A. triangularis* cysts with C1 at 1/2–1/8×MICs exhibited moderate efficiency for excystation activity with excystation rate of 23% to 28% (Fig. 9). In our study, the incomplete excystment of *A. triangularis* in PYG medium at an optimal 10 day period with the excystation rate of 100% was found (Fig. 9).

## The ability of mangostin derivative on the removal of adhesive *A. triangularis* trophozoites and cysts on contact lenses (CL)

The ability of C1 for the removal of adhesive *A. triangularis* trophozoites and cysts on contact lenses (CL) was investigated. Two commercial multi-purpose disinfecting (MPD-1 and -2) solutions and CHX were used as positive controls, while the CL without C1 exposure was considered as a negative control. Our result showed that C1 had the greatest capability to remove adherent *A. triangularis* trophozoites on CL compared to a negative control (Fig. 10A). In addition, low numbers of trophozoites and cysts adherent on CL were observed in the presence of C1, CHX, and both MPD-1 and-2 solutions (Figs. 10A and 10B). Therefore, C1 is a good candidate for CL care solution application.

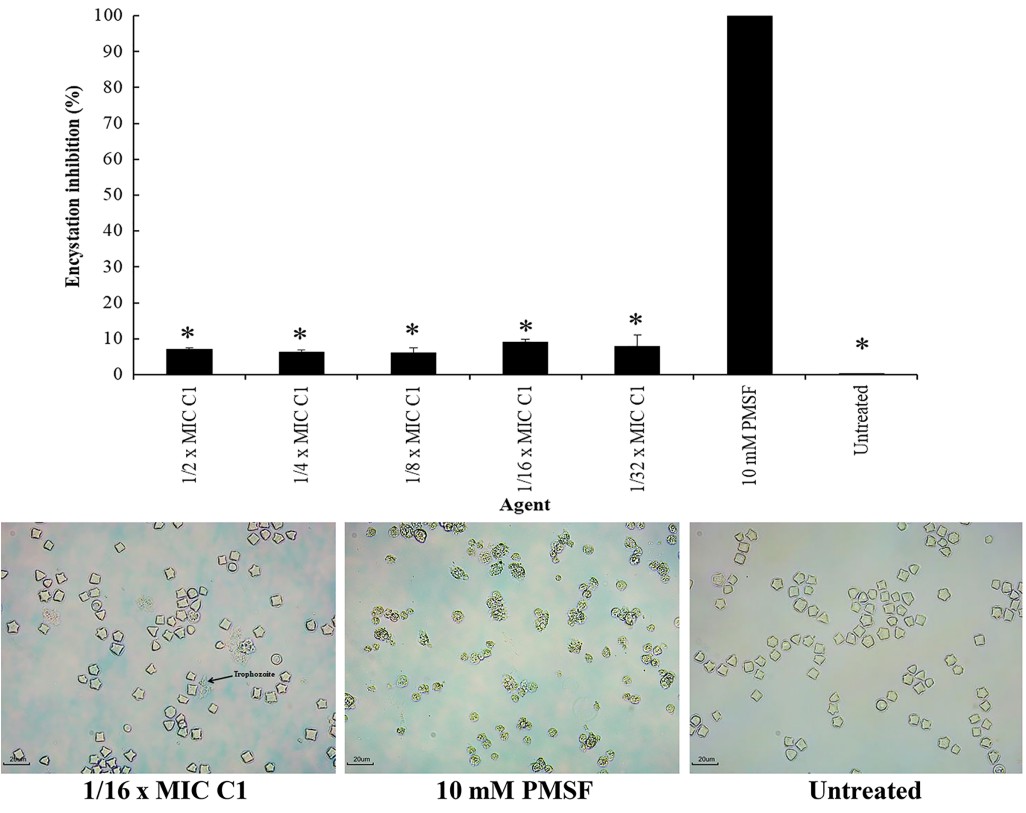

**1/16 x MIC C1**  **10 mM PMSF**  **Untreated**

**Figure 8 Inhibition of *Acanthamoeba triangularis* encystation by mangostin derivative (C1).** Inhibition encystation of *A. triangularis* after exposure to different MIC values of C1, 10 mM PMSF and without any agent (untreated; control) for 72 h. Bars represent the standard deviation (SD) ($n$ = 3). $^*p < 0.05$, statistically significant difference. Images of the encystation inhibition were observed by inverted microscope (200×) after exposure to C1 at 1/16×MIC, 10 mM PMSF, and without any agent (untreated; control) for 72 h.

## DISCUSSION

In the present study, the anti-*Acanthamoeba* of semi-synthetic mangostin derivatives was evaluated *in vitro*. Natural products are important sources for developing new drugs or supplementary products. In fact, the growing rate of *Acanthamoeba* infection causing keratitis and the toxicity of some used medicines have been reported worldwide (*Arabaci et al., 2013*; *Ortillés et al., 2017*). Therefore, the introduction of new agents is timely needed for an effective treatment. At this time, there are scanty reports on the anti-*Acanthamoeba* activity of *Garcinia mangostana extract* on *Acanthamoeba* (*Sangkanu et al., 2021*; *Sangkanu et al., 2022*). To the best of our knowledge, C1 as the semi-synthetic mangostin (epoxide) derivative has the first ever shown anti-*Acanthamoeba* activity against *A. triangularis* trophozoites (MIC and $IC_{50}$ values of 0.128 and 0.056 mg/mL, respectively); this compound also exhibited a high amoebicidal activity against cysts (MIC and $IC_{50}$ values of 0.064 and 0.035 mg/mL, respectively) proving to be the same effective as chlorhexidine (CHX). To support this, *Lorenzo-Morales et al. (2019)* reported the $IC_{50}$ values of semisynthetic substances of red alga *Laurencia viridis* named dehydrothyrsiferol against *Acanthamoeba castellanii* Neff trophozoites and cysts. Additionally, semi-synthetic

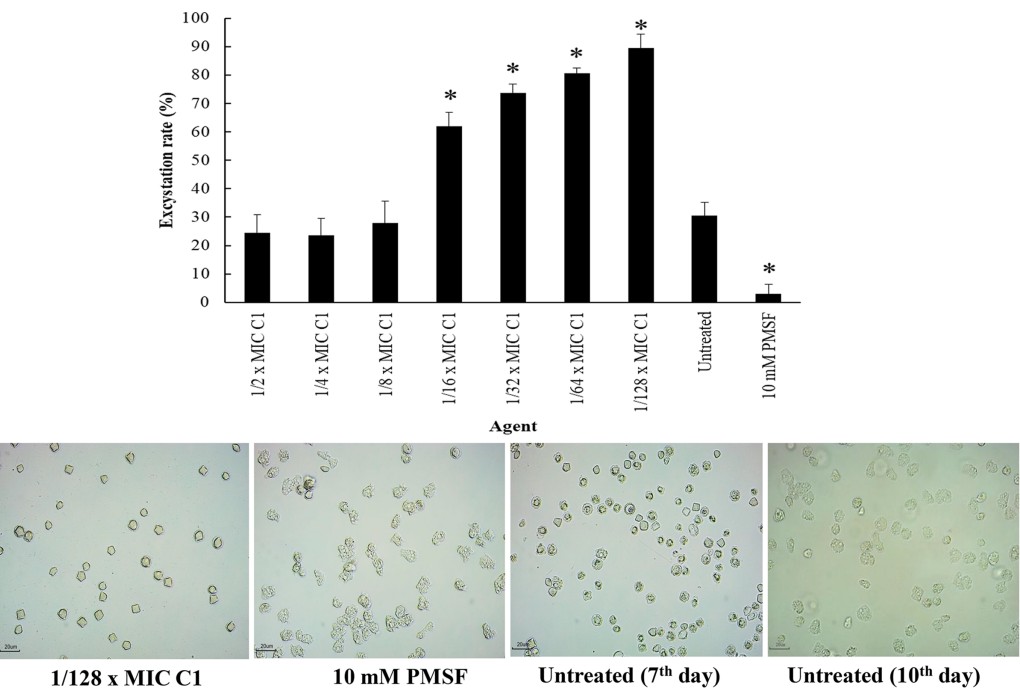

**Figure 9 Excystation activity of *Acanthamoeba triangularis* by mangostin derivative (C1).** Excystation rates of *A. triangularis* after exposure to different MIC values of C1, 10 mM PMSF, and without any agent (untreated; control) for 7 days. Values are presented as mean ± standard deviation (SD) of triplicate determinations ($n$ = 3). $^*p < 0.05$, statistically significant difference. Images of the excystation were observed by inverted microscope (200×) after exposure to C1 at 1/128×MIC, 10 mM PMSF for 7 days, and without any agent (untreated; control) for 7 and 10 days.

α-mangostin derivatives also demonstrated high potential anti-bacterial and anti-fungal activities *in vitro* (*Narasimhan et al., 2017*).

The present study clearly demonstrated the anti-*Acanthamoeba* activity of C1 supported by SEM images of morphological changes in trophozoites, including the appearance of irregular cell shapes, pore formation on the amoeba surface, and strong loss of the acanthopodia structure. Similar effects were reported after treating *Acanthamoeba polyphaga* trophozoites with cationic carbosilane dendrimers (*Heredero-Bermejo et al., 2015*). The acanthopodia are essential for host cell adhesion and cellular movements. Adhesion of *A. triangularis* to contact lens (CL) *via* acanthopodia could be one of the reasons supporting the amoeba transmission and infection in the eyes (*Ahmed Khan, 2003*). The formation of pores on the amoebic surface after treating trophozoites with C1 can lead to a leakage in the cytoplasmic content and death of the amoeba. This phenotypic change is in agreement with *Heredero-Bermejo et al. (2020)*, which demonstrated that the membrane alteration caused the *Acanthamoeba* cell death. Furthermore, C1 exhibited the anti-*Acanthamoeba* activity towards *A. triangularis* cysts. This finding was confirmed by SEM results of morphological changes, including loss of ridges, membrane blebbing, and cell shrinkage. This shrinkage might be due to the detachment of the plasma membrane from the endocyst (*Carrijo-Carvalho et al., 2017*). These morphological changes might

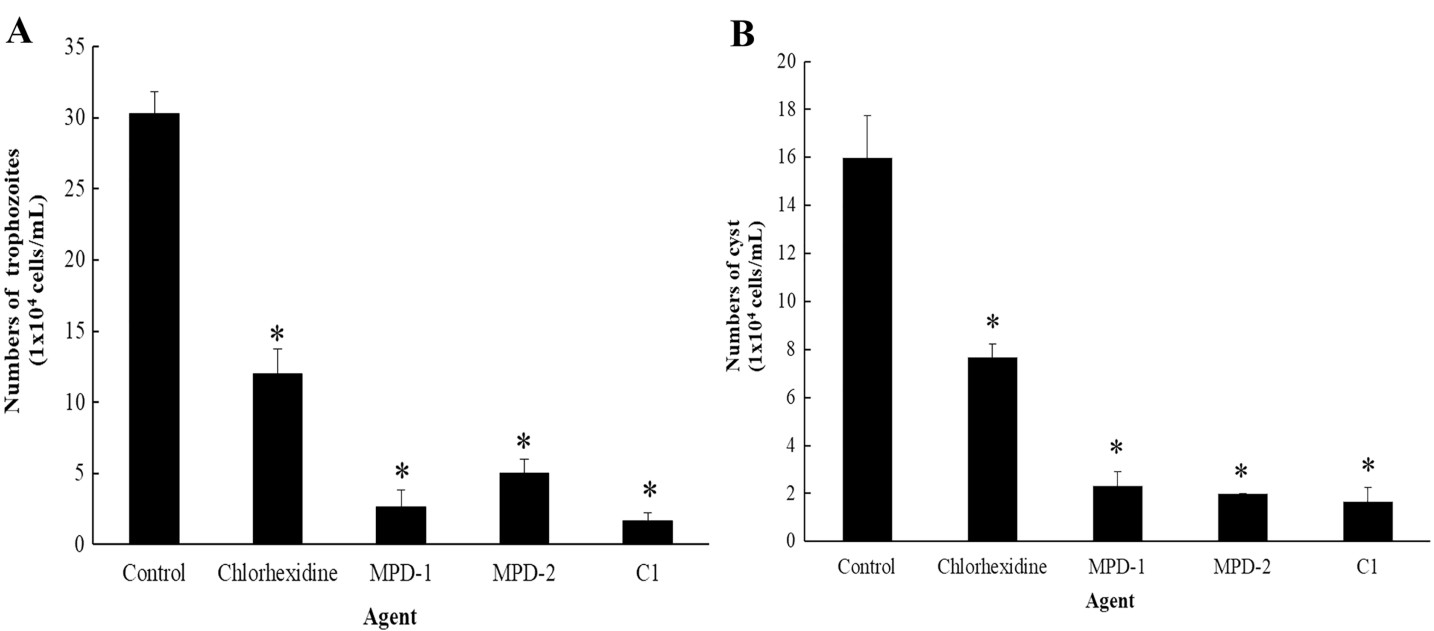

**Figure 10 The number of *Acanthamoeba triangularis* on contact lenses after treatment with chlorhexidine, MPD-1 (Renu®freshTM), MPD-2 (Duna NO RUB), and C1.** (A) Adhesive trophozoites and (B) cysts in removal test. The adhesive cells were counted using an inverted microscope. Values are presented as mean ± standard deviation (SD) of triplicate determinations ($n = 3$). $^*p < 0.05$, statistically significant difference.

imply that the cysts were destroyed. A similar observation was reported after *Acanthamoeba* spp. cysts were treated with *Aloe vera* ethanol extract and honey (*Kadry et al., 2021*).

For acridine orange and propidium iodide (AO/PI) staining, AO is an intercalating dye that can enter internal parts of *Acanthamoeba* through non-compromised membrane integrity and is able to bind to nucleic acids, emits green fluorescent detecting live and dead cells, whereas, PI can only enter dead cells with poor membrane integrity and red-orange fluorescence on the cells (*Foglieni, Meoni & Davalli, 2001*). The exposed trophozoites with C1 at 1/2×MIC value and stained with AO/PI dyes were observed as yellow-orange granules in cells. These fluorescence granules are believed to be lysosomes, taking up the AO stain through the known sequestration and digestion process of cytoplasmic macromolecules (*Kusrini et al., 2018*). *Acanthamoeba* cells treated with CHX at 1/2×MIC value of 0.008 mg/mL exhibited phenotypic characteristics of red nuclei and green cytoplasm, which indicated that they were non-viable cells. In contrast, the untreated cells were visible as green fluorescence cells, indicating healthy cells with the intact plasma membrane. The microscopy images revealed that the smaller size of the treated cells with C1 and CHX, compared to those in the untreated cells indicated the condensation of the chromatins. The similar observation in *Acanthamoeba* treated with acyclic and cyclic-samarium complexes was also described by *Kusrini et al. (2018)*. In addition, the large vacuole was found in untreated cell and *A. triangularis* cell size after exposure to C1 and CHX becomes smaller compared to untreated cells. According to the result of *Nakisah et al. (2012)*, apoptosis was found after treating *Acanthamoeba* cells with Malaysian marine

sponge *Aaptos* sp. methanol extract and stained with AO/PI technique. The condensation of cytoplasm and chromatin was established with smaller cell size. Our result indicated that exposure to the cells with C1 and CHX induced apoptosis in *Acanthamoeba*.

The present study used the calcofluor white (CFW) staining solution to distinguish between live and dead cysts. This staining solution has a specific affinity for cellulose, a major component of the *Acanthamoeba* cyst wall (*Marines, Osato & Font, 1987*). It indicated that the cysts treated with C1 lose the CFW staining due to loss or leakage of the cell wall, leading ultimately to cell death. This leakage may emanate from conformational changes and weakening of the inter-fiber bonds of the cellulose cell wall (*Mooney et al., 2020*). In addition, the cytotoxicity assay demonstrated that C1 was non-cytotoxic on Vero cells, whereas CHX displayed a strong cytotoxic effect. CHX was found to be a cytotoxic agent to human osteoblastic cell line (*Lee et al., 2010*), human dermal fibroblasts (*Hidalgo & Dominguez, 2001*), and human alveolar bone cells (*Cabral & Fernandes, 2007*). Our results agree with previous research in which CHX exhibited cytotoxic effect on Vero cells (*Baqui et al., 2001*).

Interestingly, it was found that C1 showed an effect on encystation inhibition. Encystation of *Acanthamoeba* is the phenotypic transformation from vegetative trophozoites to cysts which occurs under unfavorable conditions (*Chávez-Munguía et al., 2005*). C1 gave the encystation inhibition of 9%, while it was not detected in the negative control (without adding C1). In addition, C1 was effective in increasing excystation rate of up to 85% compared to the negative control. In literature, the process of excystment consists of five stages: mature cyst, activated cyst stage, early pre-emergence stage, late pre-emergence, and complete emergence (*Mattar & Byers, 1971*). Excystment occurs when mature cysts are transferred from the encystment medium (Neff's encystment medium) back into the growth medium that contains proteose peptone, yeast extract and glucose (PYG medium). However, the time of *Acanthamoeba* excystment processes depends upon species or conditions (*e.g.*, pH, temperature, and time duration) (*Mattar & Byers, 1971*; *Marciano-Cabral & Cabral, 2003*). In our study, excystment of *A. triangularis* was completed within 10 days in PYG medium with the excystation rate of 100%. Many proteases (cysteine, serine, and metal) are used for encystation and excystation of protozoan parasites, including *Acanthamoeba* (*Lee et al., 2015*). PMSF is a serine protease inhibitor that inhibits serine protease activity; therefore, encystation and excystation were diminished in the presence of this inhibitor. According to the result, C1 may stimulate the protease activity that leads to a high excystation rate in the presence of C1 compared to the control. Generally, *Acanthamoeba* trophozoites transform into a cyst form to protect themselves under harsh conditions such as extreme temperatures, extreme pHs, and anti-agents. In our study, we found that *Acanthamoeba* maintained themselves into cyst form under high concentrations of C1 (1/2 to 1/8×MIC values) for survival against stress conditions; therefore excystation rates were low. Whereas, at low concentrations of C1 (1/16 to 1/128×MIC values), *Acanthamoeba* cysts transformed into trophozoites because the excystation activity was induced. Based on our results, the concentrations of C1 can therefore be divided into three categories: (1) MIC values for cyst destruction, (2) 1/2 to 1/8×MIC values for inhibition of excystation activity and (3) 1/16 to 1/128×MIC values for

induction of excystation activity. These C1 concentrations were of paramount importance for the development of future therapeutic agents of *Acanthamoeba* infections.

In this study, we further found that C1 showed high efficiency in removing adhesive *A. triangularis* trophozoites and cysts from CL surface. This may be due to a lack of acanthopodia which is the main adhesive structure of trophozoites used for the surface attachment such as contact lenses (*Sangkanu et al., 2022*). Moreover, *Acanthamoeba* cells change to irregularly shaped cells and rounded trophozoites in response to the compound in order to protect themselves from environmental conditions such as stress (*Boonhok et al., 2021*). Based on this experiment, C1 is considered as a potential candidate for future studies mainly to demonstrate the actual role of this compound against other pathogenic *Acanthamoeba* infections and to explore the potential pharmaceutical applications in the future. Furthermore, the results obtained in this study revealed that C1 showed high efficacy in removing adhesive *A. triangularis* trophozoites and cysts from CL surface, which can be applied for lens care solution. In addition, an assessment of additional pathogenic clinical strains of *Acanthamoeba* sp. (*e.g.*, *A. castellanii* and *A. polyphaga*) is of paramount importance in our future studies.

## CONCLUSIONS

Our study indicated that the mangostin derivative named 1,3-Dihydroxy-6-(2,3-epoxypropoxy)-7-methoxy-2,8-bis(3-methylbut-2-enyl)-9H-xanthen-9-one (C1, epoxide derivative) exhibited the potent anti-*Acanthamoeba* activities against *A. triangularis* trophozoites and cysts. This compound showed ability in encystation inhibition and induced the *Acanthamoeba* excystation activity. Furthermore, it is interesting to note that C1 exhibited high potential on the removal of adhesive *A. triangularis* trophozoites and cysts from contact lenses (CL). Our result suggests that C1 could be a potential source in the development of cleaning or disinfecting solutions for contact lenses and a novel therapeutic agent against *Acanthamoeba* infections in the future.

### Funding

This work was supported by the the Royal Patronage of Her Royal Highness Princess Maha Chakri Sirindhorn—Botanical Garden of Walailak University, Nakhon Si Thammarat, under the project entitled: Medicinal Thai Native Plants against *Acanthamoeba triangularis* as a serious eye infection (WUBG 031-2565) and project CICECO-Aveiro Institute of Materials, UIDB/50011/2020, UIDP/50011/2020 & LA/P/0006/2020, FCT/MEC (PIDDAC). The funders had no role in study design, data collection and analysis, decision to publish, or preparation of the manuscript.

### Grant Disclosures

The following grant information was disclosed by the authors:
Botanical Garden of Walailak University: WUBG 031-2565, UIDB/50011/2020, UIDP/50011/2020 & LA/P/0006/2020, FCT/MEC (PIDDAC).

## Competing Interests

The authors declare that they have no competing interests.

## Author Contributions

- Julalak Chuprom conceived and designed the experiments, performed the experiments, analyzed the data, prepared figures and/or tables, authored or reviewed drafts of the article, and approved the final draft.
- Suthinee Sangkanu conceived and designed the experiments, performed the experiments, analyzed the data, prepared figures and/or tables, authored or reviewed drafts of the article, and approved the final draft.
- Watcharapong Mitsuwan conceived and designed the experiments, performed the experiments, authored or reviewed drafts of the article, and approved the final draft.
- Rachasak Boonhok conceived and designed the experiments, authored or reviewed drafts of the article, and approved the final draft.
- Wilawan Mahabusarakam conceived and designed the experiments, performed the experiments, authored or reviewed drafts of the article, and approved the final draft.
- L. Ravithej Singh conceived and designed the experiments, prepared figures and/or tables, authored or reviewed drafts of the article, and approved the final draft.
- Ekachai Dumkliang conceived and designed the experiments, prepared figures and/or tables, authored or reviewed drafts of the article, and approved the final draft.
- Kritamorn Jitrangsri conceived and designed the experiments, prepared figures and/or tables, authored or reviewed drafts of the article, and approved the final draft.
- Alok K. Paul conceived and designed the experiments, analyzed the data, prepared figures and/or tables, authored or reviewed drafts of the article, and approved the final draft.
- Sirirat Surinkaew conceived and designed the experiments, performed the experiments, prepared figures and/or tables, and approved the final draft.
- Polrat Wilairatana conceived and designed the experiments, authored or reviewed drafts of the article, and approved the final draft.
- Maria de Lourdes Pereira conceived and designed the experiments, analyzed the data, authored or reviewed drafts of the article, and approved the final draft.
- Mohammed Rahmatullah conceived and designed the experiments, authored or reviewed drafts of the article, and approved the final draft.
- Christophe Wiart conceived and designed the experiments, authored or reviewed drafts of the article, and approved the final draft.
- Sonia Marlene Rodrigues Oliveira conceived and designed the experiments, analyzed the data, authored or reviewed drafts of the article, and approved the final draft.
- Veeranoot Nissapatorn conceived and designed the experiments, analyzed the data, authored or reviewed drafts of the article, and approved the final draft.

## Data Availability

The raw measurements are available as a Supplemental File.

## Supplemental Information

Supplemental information for this article can be found online at http://dx.doi.org/10.7717/peerj.14468#supplemental-information.

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
