# Peer review of "Anti-Acanthamoeba activity of a semi-synthetic mangostin derivative and its ability in removal of Acanthamoeba triangularis WU19001 on contact lens"

_PeerJ, doi:10.7717/peerj.14468_

## Round 0.1 · original submission · Major Revisions

The review process is now complete, and three reviews are included at the bottom of this letter (also see the annotated file). All reviewers and I agree that your manuscript deserves to be published. While the experimental design and findings are original and relevant to the field, we identified some concerns and flaws that must be rigorously considered in your resubmission.

All sections of the manuscript (including references) need to be carefully revised and modified accordingly. Results must be discussed deeply. English language must be corrected and improved to make the manuscript clearer to the readers.

Reviewer 1 ·

Basic reporting

The manuscript addresses an unmet medical need in drug discovery targeting Acanthamoeba infections, including Acanthamoeba keratitis that affects in large extend compact lens wearers. The identification of the plant natural product mangostin and its derivatives as anti-amoebic agents is an interesting advancement toward new therapeutic options for these infections.

The manuscript has a number of drawbacks that need to be address prior to publication. English needs substantial improvement throughput and particularly in Discussion that is poorly written. In addition, paragraphs duplicating Introduction should be removed from the Discussion section and multiple redundancies should also be eliminated from the text.

Manuscript title must be modified. It is incorrect grammatically and also incorrectly references mangostin derivatives as "synthetic compounds". These derivatives are not synthetic, as they were obtained by derivatization of mangostin isolated from the fruits. Semi-synthetic would be more correct definition.

The phrase "synthetic compounds from mangostin modification" should be replaces with "semi-synthetic mangostin derivatives" on the first usage and "mangostin derivatives" further down throughout the text.

The rational to selecting epoxy- and nitrile- mangostin derivatives is not mentioned in the manuscript. Why these specific derivatives were explored?

Materials and methods section does not require a figure. Figure 1 should be removed from the Materials and Methods. It would be more appropriate as a table of content graphics (TOC) or perhaps Introduction. Minor comment: Figure 1 does not show any synthesis as it is stated in the text.

Are there morphological differences between the cells treated with compound 1 and CHX that may provide a clue on mechanism of action or potential target?

Lane 146: "Briefly, the air‐dried powdered pericarps of G. mangostana (4 kg) were soaked in 2.0 L of dichloromethane (CH2Cl2) for 48 h" - specify "pericarps of G. mangostana fruits"

Lane 192: "...treated with sodium dodecyl sulfate (SDS) solution at 0.5% (w/v) final concentration for 20 min to solubilize trophozoites and immature cysts while mature cysts were resistant to SDS solution" - did you or others check if 0.5% SDS affects mature cyst susceptibility to drugs?

Lanes 234-250: Anti-cyst, anti-cyst formation, anti-encystation rate - is incorrect terminology. There is no such things like anti-cyst or anti-encystation. Reformulate.

Lane 267: "Effects of synthetic compound from mangostin modification on the removal of adhesive Acanthamoeba trophozoites and cysts on contact lenses (CL)" - change to "Effect of mangostin derivatives on the adhesion of Acanthamoeba to the surface of contact lenses (CL)"

Lane 530: "this compound may result in the induction of cysteine protease which is important for process of Acanthamoeba excystation" - are there any evidence for this assumption?

More notes are provided in the attached PDF file.

Experimental design

No comment

Validity of the findings

No comments

Annotated reviews are not available for download in order to protect the identity of reviewers who chose to remain anonymous.

Reviewer 2 ·

Basic reporting

The ms presents the activity of a previously studied source by the same authors, some extra data are added to that previous work.
The data are of interest however a colorimetric assay i.e Alamar Blue should have been used for the ICs calculation.

Experimental design

A colorimetric assay should be performed. A rationale not to use should be included.

Validity of the findings

the obtained data are of interest

Additional comments

Please improve magnification and quality of images of amoebae and cysts in figs 6-8

Reviewer 3 ·

Basic reporting

Anti-Acanthamoeba activity and its ability in removal of Acanthamoeba triangularis WU19001 genotype T4 on contact lens of synthetic compounds from mangostin modification (#72271)
In this study the authors tested the efficacy of three semi-synthetic compounds derived from mangostin against throphozoites and cysts of an Acanthamoeba strain. Furthermore, they evaluated the impact of the most effective of these three compounds on encystment, excystment, adhesion to contact lenses and its cytotoxicity towards Vero cells. Based on their findings they propose a potential therapeutic applicability in Acanthamoeba therapy.

General remarks.
I am not a native English speaker and I am sympathetic to a certain extent, but the quality of the English language in the manuscript should be improved, in particular the discussion section. The title of the study is symptomatic for the entire manuscript. Sometimes it makes it harder to understand, what the authors want to say. While obviously many scientists participated in this study and worked hard in the laboratory, the proof-reading part was neglected. The manuscript should be carefully checked by a fluent speaker and revised accordingly.

In my opinion the manuscript could be shorter and more concise, also taking into account that the experiments are more or less a second serving of the papers from Sangkanu et al. 2021 and 2022, with the crude extract.

The experiments in the Materials and Methods section are in a different order than in the results section. Also figures are not in the same order, this should be consistent.
Some citations are missing in the Reference section and some should be replaced by more appropriate ones. In my opinion the researchers who established the facts should be cited, if possible. With some citations I had the feeling that the authors cited papers, that just mention the information, but did not contribute to their establishment. Of course, citing review papers is quite common, but could be avoided at least at some occasions in the manuscript.

Altogether I strongly advise the authors to pay more attention when proof-reading their manuscript and with their citations. In times like this when papers get rejected because of small things due to the overload of manuscripts, these flaws could be avoided. Furthermore, it makes the reviewing process a lot more work.

Some findings in this study are rather unusual, for instance that the compound was more effective against cysts and did inhibit encystment and excystment at the same time. This could be worked out in a little more detail in the discussion, since it is rather peculiar.

Altogether, the authors present an interesting study, tried to investigate several aspects of the interaction of their compound and the amoebae and provide a lot of information. The methodology is sound and the results are convincing. However, since the manuscript in its current form is not suitable for Acceptance and should be carefully revised and rewritten and I would like to have some additional information (see questions in “specific remarks”) I recommend “Major revision”.

Experimental design

no comment

Validity of the findings

no comment

Additional comments

Specific Remarks
Synthetic compound from mangostin modification (compound 1) is used throughout the entire manuscript. Maybe there is a shorter option or abbreviation for it!
The same I would suggest for “Acanthamoeba triangularis WU19001 genotype T4”, it is very long and could be abbreviated.
Line 78ff: I would rephrase this paragraph. Infections due to CL wear can lead to several problems as the authors mention, but together with the next sentence it sounds funny. “These diseases …..”, its not diseases as such. And they are not commonly caused by Acanthamoeba. AK is still rare in contrast to other keratitis infections causes by bacteria or fungi.
Ref: Hahn et al. The reference is not the most appropriate one for the general introduction of AK.
There is no reference for Staphylococcus, but for Pseudomonas.
Line 87: again the Ref. of Koyun et al. is not the most appropriate.
Line 96 Ref. not appropriate
Line 101. Ref not appropriate, maybe Bergmanson et al. but not Lonnen et al.
Line 102ff: This paragraph is not well phrased and should be revised.
Line 109: Ref. Fanselow et al. not in Ref. section
Line 110: Chu et al. not appropriate
Line 116: Ref. not appropriate unless the authors rephrase the sentence.
Line 120: in literature… This group published two studies already on this topic, which should be mentioned and cited here.
Line 122: the rest of this paragraph should be re-written.
Line 180: The medium in deLeo and Baveye is different. Provide another reference or just list the ingredients.
Line 197: How can you determine the viability of cysts with trypan blue? In my experience cyst always appear dark blue with trypan blue, not matter if they are dead or alive, or is that wrong?
Line 231: What does inhibit mean? Are the amoebae dead? Is this the minimal effective concentration to kill 90% of the trophozoites. Inhibition of amoebae is hard to define, since movement cannot really be taken as an indicator. But staining of course can, and so it should be stated, whether these 90% of amoebae are dead.
Line 234 anti-encyst formation sounds funny! This is up to the authors but when I read the manuscript I found the term anti-encystation irritating. For me “inhibition of encystment” or something like that would be more comprehensible. Just a suggestion!
Line 255: Reference is not provided in Ref.section
Line 261: is it really possible to determine the rate of excystation after 7 days? Excystation does not require so much time as far as I know. After seven days there would be a lot more amoebae than just the trophozoites that hatched from the cysts?
Line 308: Please provide concentration of compound used for this experiment.
Line 321: Please provide concentration of compound used for this experiment.
Line 333: Please provide concentration of compound used for this experiment
Line 374: was it 0.128 or half of it as described in the figure legend?
Line 282: again, 0.064 or half?
Line 403: Ref. not appropriate.
422: Why would the untreated cysts not show a higher excystation rate after 7 days? Shouldn’t this rate be a lot higher?
Line 447ff: I am not sure whether I understand it right, but the concentration that is tolerated by Vero cells is 0.008 – 0.030mg/ml with 0.030 mg/ml being the IC50. The MIC for trophozoites is 0.128 mg/ml with 90% of the amoebae “inhibited”. How can I compare these results? For me it sounds like the amoebae tolerate a lot more than the Vero cells, which would not be a positive result. However, the concentrations cannot be directly compared, since its IC50 and IC90 (or is it not?). I confess that I don’t have so much practical experience with this kind of experiments, however, for me it sounds like the compound is not suitable for an therapeutic approach due a low selectivity index? Maybe it is still promising for disinfection or lens care but not for application onto the eye.
Line 464: Ref. is missing in Ref.section
Line 484: Ref. is missing in Ref. section
Line 492f: Please rephrase.
Line 496: Ref. not appropriate
Line 500f: The exposure of…. Please rephrase!
Line 502ff: This passage needs revision!
Line 517: Reference. Moon et al. made a nice study, but were not the first to establish that Acanthamoeba cysts consist of cellulose. Please provide another reference.
Line 523: Reference is not appropriate.
Line 528: Hong et al. reported on a cysteine protease involved in encystment, but of the cathepsin L family, inhibited by E-64. PMSF is inhibiting serine proteases and the cysteine protease papain, and serine proteases were also shown to be involved in the excystment process, that might be the reason why PMSF is inhibiting excystment. How can you propose an induction for cysteine proteases? Is there any evidence in literature? (and I am not saying, you are wrong, of course that might be possible!)
Line 531: This statement is a little unsuitable for your study, since the tested compound was more effective against cysts. It is even more effective against cysts than trophozoites, but at the same time induces excystment? Of course in a different concentration, however, that raises more questions!
Line 552: I would suggest a more positive conclusion, rather than limitations you could call it challenges for the future.
Reference:
Darzynkiewicz et al. seems to be missing in the text.
Figures:
Figure 1 is nice to look at, however, if space is limited, this figure could be omitted.
Figure 7: I am wondering whether the results for the compound are truly significant compared to PMSF? Rather than letters, asterisk should be used with the p-value ranges provided in legends. I am not sure, whether the photos are necessary.
Figure 8: Again letter are confusing for p-values, maybe asterisk would be clearer. Photos maybe not necessary.
Figure 9: Renu and Duna are the contact lens solutions, I guess! Please provide this info in the MM section and Figure legend.

---

## Round 0.2 · Major Revisions

Although most of the changes suggested by the reviewers have been made, the manuscript still needs revision. The abstract and Discussion sections must be revised aiming to properly address the main results obtained. English language must be corrected and improved. Please, note that this requirement was pointed out by two reviewers and corroborated by me.

Reviewer 1 ·

Basic reporting

The revised version of the manuscript is submitted. The authors carefully addressed the Reviewer’s comments and improved the manuscript. English language is also somewhat improved; however, help from a colleague proficient in English and familiar with the subject matter is still needed. Some examples of needed improvements (underlined) are provided below.

Another recommendation is to modify the Abstract and Discussion to more accurately communicate and summarize the experimental results. Thus, the authors cannot claim that compound 1 “can solve resistance problems of cyst related to a poor response to medical therapy” (lines 63-64). Toxicity of compound 1 tested in kidney Vero cells (0.030 mg/ml) and its anti-Acanthamoeba activity (0.056 and 0.035 mg/mL for trophozoite and cyst, respectively) both are in the same range that may render compound unsuitable for application onto the eye. It would be useful to compare toxicity of 1 with chlorhexidine (CHX) in the same Vero cells assay. CHX is quite toxic, so compound 1 may have an advantage in this regard.

Likewise, the statement “mangostin derivative could be one of the future candidate drugs in Acanthamoeba treatment” (lines 67-68) is not supported by experimental data. Smilar statement in Discussion (lines 561-563) should also be removed.

Although anti-cyst activity of 1 is of significant interest, acquiring additional toxicity data using more relevant tissues and cells would be required to further examine compound 1 potential for the treatment of eye infection. Compound 1 may still be promising for lens disinfection. Abstract (lines 63-69) has to be re-written to adequately address the significance of finding.

Examples of other improvements:

Title
“Anti-Acanthamoeba activity of semi-synthetic mangostin derivatives and its ability in removal of
Acanthamoeba triangularis WU19001 on contact lens” - mixed up plural and singular usage. Correct to ‘derivative and its’ or ‘derivatives and their’.

Abstract
48-49: Garcinia mangostana L. or mangosteen tree is a well-known medicinal plant native to Southeast Asia producing a great variety of pharmacologically active compounds, including xanthonoid mangostin.
49-52: Our study examined pharmacological activities of semi-synthetic mangostin derivatives, that is, amoebicidal activity, encystation inhibition, excystation activity, and removal adhesion of Acanthamoeba on contact lenses (CL).”
54-66: Phenotypically, the SEM images showed the strong loss of acanthopodia, pore formation in the cell membrane, and membrane damage in the trophozoites exposed to 1, while the treated cyst shrunk and adopted an irregular flat cyst shape.
The mangostin derivative (compound 1) = 1. You may reference derivatives by numbers once you defined them at the begining.
58-59: …induced condensation of cytoplasm and chromatin with cell shrinkage.
Shrinkage = smaller cell size
61-62: In addition, in the presence of 1, A. triangularis excystation was demonstrated…
To be concise, reference compounds by numbers (instead of ‘this mangostin derivative’ or ‘compound 1’) throughout the text.

Introduction
78-82: A wide range of microorganisms can cause IK, for example, filamentous fungi (Fusarium spp. and Aspergillus spp.), bacteria (Pseudomonas aeruginosa, Staphylococcus aureus, Streptococci spp., Enterobacteriaceae spp., Corynebacterium spp., and Propionibacterium spp.), viruses, and protozoa (Acanthamoeba spp.) (OíCallaghan et al., 2019; Lee et al., 2021; Ting et al., 2021).

101-102: The most routinely used anti-Acanthamoeba keratitis drugs are antiseptics chlorhexidine (CHX) and polyhexamethylene biguanide (PHMB).

105-106: ‘However, the potential toxicity of PHMB on human corneal cells has been recently reported (Lorenzo-Morales 106 et al., 2013)’ – in 2022, “recently” cannot be applied to something happened in 2013.

107-110: “Plants are considered a rich source of potential therapeutic agents which have been traditionally used in the treatment of several diseases. Also, there are sources of amoebicidal agents against Acanthamoeba infection with high anti-amoebic activity and low toxicity (Niyyati, Dodangeh & Lorenzo-Morales, 2016) – consider revision. These two sentences do not belong together.

112-114: “Mangosteen pericarp (MP) is an important source of bioactive xanthones, which has remarkable antioxidant, anti-inflammatory, immunomodulatory, antiviral, antifungal, antibacterial, and anticancer activities (Abate et al., 2022; Nauman & Johnson, 2022)” – mixed-up plural and singular usage. Should be: xanthones, which have…

121-123: “The pharmacological activities of these compounds named 1,3-dihydroxy-6-(2,3-epoxypropoxy)-7-methoxy-2,8-bis(3-methybut-2-enyl)-9H xanthen-9-one were investigated” – if it is one compound, use singular: this compound named … was investigated.
Is it compound 1? If so, you need to indicate that once in the paper and then consistently use abbreviation 1 to refer this compound.

Materials & Methods
129-130: “The synthesis and efficacy testing procedures of synthetic compounds from mangostin modification are summarized in Fig. 1” – there are no synthesis or testing procedures shown in Fig. 1. Correct to ‘mangostin modifications’ because more than one modifications are shown in the figure.

245, 278, 294, 314, 330, 346 and many other places throughout the text:
Replace ‘mangostin derivative (compound 1)’ or compound 1 with 1. You have already defined it earlier.

Results
On the same page, two different sets of IC50 for trophozoites and cysts are provided:
374-375: compound 1 showed greater anti-Acanthamoeba activity at 0.128 mg/mL and 0.064 mg/mL on trophozoites and cysts, respectively (Table 1)
381-382: Compound 1 demonstrated the inhibitory activity against the Acanthamoeba trophozoites with IC50 of 0.05621 mg/mL, while the cyst IC50 was 0.03522 mg/mL
Figure 2 also refers IC50 of 0.05621 mg/mL for trophozoites, and 0.03522 mg/mL for cyst.
However, although throughout the text, the authors refer to 0.128 mg/mL as IC50 for trophozoites and 0.064 mg/mL as IC50 for cyst (line 387 and further down the text). You need to stick to one set of values and eliminate (or explain) the IC50 discrepancies.

Discussion
Discussion is excessively long due to repeating results and providing additional details. For instance, lines 489-517 largely describe SEM results. This section should be substantially reduced and a concise message pertinent to discussion should be presented instead.

Discussion should be divided into paragraphs, each addressing a specific property of the compounds. Otherwise, Discussion looks like a laundry list where one topic merges to the next one without a clear-cut message.

Finally yet importantly, Discussion needs serious editing by a colleague proficient in English and familiar with the subject matter. The authors may also consider help of professional editing service to make discussion more readable. Some sentences have to be read 3-5 times to be understood.

A few examples of incorrectly composed sentences include (but are not limited to):

480-481: While, fluorescent dyes are reportedly sensitive to light which should be carefully used.

519-520: The similar observations in Acanthamoeba treated with acyclic and cyclic-samarium complexes as also described by Kusrini et al. (2018).

523-525: According to Nakisah et al. (2012), apoptosis was evaluated morphologically after the treated Acanthamoeba cells with Malaysian marine sponge Aaptos sp. methanol extract and stained with AO/PI technique.

Experimental design

It would be useful to compare toxicity of compound 1 with chlorhexidine (CHX) in the same Vero cells assay. CHX is quite toxic, so compound 1 may have an advantage in this regard.

Validity of the findings

The authors have to re-write Abstract (lines 63-69) to more accurately summarize the experimental results and adequately address the significance of finding.

Annotated reviews are not available for download in order to protect the identity of reviewers who chose to remain anonymous.

Reviewer 2 ·

Basic reporting

Authors have clearly enhanced the document

Experimental design

It is ok

Validity of the findings

The data are of some interest

Reviewer 3 ·

Basic reporting

no comment

Experimental design

no comment

Validity of the findings

no comment

Additional comments

In general all remarks were considered and the authors tried to answer all questions and put a lot of effort into doing so.
However, the mansucript is still poorly written and the English needs improvement. The authors should have a fluent speaker to revise the manuscript.
It is not suitable for publication in its current form.

---

## Round 0.3 · Minor Revisions

The authors improved the manuscript significantly. Please, discuss more properly the effect of C1 concentration on the Acanthamoeba excystation activity.

Reviewer 1 ·

Basic reporting

The manuscript in its second revision substantially improved. Minor language corrections (as indicated below) are required.

Clarity of result reporting needs improvement:

Concerning the encystation activity, the encystation inhibition of C1 does not exceed 10% at any C1 concentrations compared to untreated control (Figure 8). If would be fair if the authors stated that more specifically where appropriate. In Discussion (lane 545), instead of “C1 showed the greater encystation inhibition than the negative control" use "9% inhibition compared to negative control". Also in the Abstract (lane 64-65), instead of “displayed the highest inhibitory effect on A. triangularis encystation at 1/16xMIC value (0.004 mg/mL)” use “displayed 9% inhibitory effect…” Otherwise, this result remains undisclosed until the very end of the manuscript.

Same about “excystation activity” – activity can be anything. You have to specifically state in the Abstract (lane 66) that excystation was induced by lower C1 concentrations reaching 89.5% at 1/128xMIC.
Do you have an explanation why excystation rate is largely unaffected by higher C1 concentrations (1/2xMIC -1/8xMIC) and is induced by lower C1 concentrations (1/16xMIC-1/128xMIC) compared to untreated control (Figure 9)? How this effect may be beneficial (or detrimental) for curative effect of the drug? This should be addressed in discussion.

Suggested corrections:
Lane 145: correct to “The extraction and isolation procedures of mangostin were previously reported by…”
Lanes 268-269: correct to “the supernatant was discarded and the pellet washed twice with…”
Lanes 434-436: correct to “According to ISO 10993-5, at the cell viability exceeding 80%, compound is considered non-cytotoxic, while at the viability range between 60-80 %, as weakly cytotoxic.”
Lane 437: correct to “the weakly cytotoxic concentration of C1 was 0.08 mg/mL”
Lanes 437-438: correct to “Therefore, the results confirmed that C1 had no cytotoxic effect on Vero cells at inhibitory concentrations.”
Lane 440: correct to “CHX had strong cytotoxic effect on Vero cells at inhibitory concentrations”

Experimental design

no comments

Validity of the findings

no comments

Reviewer 3 ·

Basic reporting

no comment

Experimental design

no comment

Validity of the findings

no comment

Additional comments

The authors improved the manuscript accordingly. It is suitable for publication after a final proof-reading.

---

## Round 0.4 · Minor Revisions

The Section Editor identified some changes that are needed:

? In the section of in vitro cell citotoxicity (lines 434-441), the text is contradictory "The results revealed that C1 displayed strong cytotoxicity against Vero cells showing only 3.94% of cell viability at a concentration of 0.16 mg/mL, while CHX demonstrated strong cytotoxicity at the concentration above 0.01 mg/mL (Fig. 7). The IC50 values of C1 for Acanthamoeba trophozoites and cysts were 0.035 and 0.056 mg/mL, respectively. According to ISO 10993-5, at the cell viability exceeds 80%, compound is considered non-cytotoxic, while at cell viability range between 60-80%, as weakly cytotoxic. In this study, the non-cytotoxic dose of C1 was 0.04 mg/mL and the weakly cytotoxic concentration of C1 was 0.08 mg/mL. Therefore, the results confirmed that C1 had no cytotoxic effect on Vero cells at inhibitory concentrations."

> I do not dispute the conclusion regarding the non-cytotoxicity, but rather the way that the beginning of the section claims "strong cytotoxicity" and the end of the section walks back from this claim and indeed correctly concludes the exact opposite.

---

## Round 0.5 · accepted · Accept

The authors have satisfactorily responded to the points raised and made the necessary changes to the manuscript.